# Noise-Tolerant Interactive Learning Using Pairwise Comparisons

Yichong Xu[*], Hongyang Zhang[*], Kyle Miller[†], Aarti Singh[*], and Artur Dubrawski[†]

[*]Machine Learning Department, Carnegie Mellon University, USA
[†]Auton Lab, Carnegie Mellon University, USA
{yichongx, hongyanz, aarti, awd}@cs.cmu.edu,
mille856@andrew.cmu.edu

## Abstract

We study the problem of interactively learning a binary classifier using noisy labeling and pairwise comparison oracles, where the comparison oracle answers which one in the given two instances is more likely to be positive. Learning from such oracles has multiple applications where obtaining direct labels is harder but pairwise comparisons are easier, and the algorithm can leverage both types of oracles. In this paper, we attempt to characterize how the access to an easier comparison oracle helps in improving the label and total query complexity. We show that the comparison oracle reduces the learning problem to that of learning a threshold function. We then present an algorithm that interactively queries the label and comparison oracles and we characterize its query complexity under Tsybakov and adversarial noise conditions for the comparison and labeling oracles. Our lower bounds show that our label and total query complexity is almost optimal.

## 1 Introduction

Given high costs of obtaining labels for big datasets, interactive learning is gaining popularity in both practice and theory of machine learning. On the practical side, there has been an increasing interest in designing algorithms capable of engaging domain experts in two-way queries to facilitate more accurate and more effort-efficient learning systems (c.f. [26, 31]). On the theoretical side, study of interactive learning has led to significant advances such as exponential improvement of query complexity over passive learning under certain conditions (c.f. [5, 6, 7, 15, 19, 27]). While most of these approaches to interactive learning fix the form of an oracle, e.g., the labeling oracle, and explore the best way of querying, recent work allows for multiple diverse forms of oracles [12, 13, 16, 33]. The focus of this paper is on this latter setting, also known as active dual supervision [4]. We investigate how to recover a hypothesis $h$ that is a good approximator of the optimal classifier $h^*$, in terms of expected 0/1 error $\Pr_X[h(X) \neq h^*(X)]$, given limited access to labels on individual instances $X \in \mathcal{X}$ and pairwise comparisons about which one of two given instances is more likely to belong to the +1/-1 class.

Our study is motivated by important applications where comparisons are easier to obtain than labels, and the algorithm can leverage both types of oracles to improve label and total query complexity. For example, in material design, synthesizing materials for specific conditions requires expensive experimentation, but with an appropriate algorithm we can leverage expertize of material scientists, for whom it may be hard to accurately assess the resulting material properties, but who can quickly compare different input conditions and suggest which ones are more promising. Similarly, in clinical settings, precise assessment of each individual patient's health status can be difficult, expensive and/or risky (e.g. it may require application of invasive sensors or diagnostic surgeries), but comparing relative statuses of two patients at a time may be relatively easy and accurate. In both these scenarios we may have access to a modest amount of individually labeled data, but the bulk of more accessible training information is available via pairwise comparisons. There are many other examples where

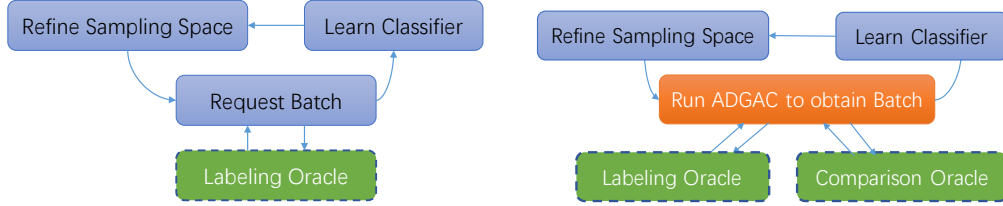

Figure 1: Explanation of work flow of ADGAC-based algorithms. **Left:** Procedure of typical active learning algorithms. **Right:** Procedure of our proposed ADGAC-based interactive learning algorithm which has access to both pairwise comparison and labeling oracles.

Table 1: Comparison of various methods for learning of generic hypothesis class (Omitting $\log(1/\varepsilon)$ factors).

| Label Noise | Work | # Label | # Query | $\mathsf{Tol}_{\text{comp}}$ |
|---|---|---|---|---|
| Tsybakov ($\kappa$) | [18] | $\tilde{\mathcal{O}}\left(\left(\frac{1}{\varepsilon}\right)^{2\kappa-2}d\theta\right)$ | $\tilde{\mathcal{O}}\left(\left(\frac{1}{\varepsilon}\right)^{2\kappa-2}d\theta\right)$ | N/A |
| Tsybakov ($\kappa$) | **Ours** | $\tilde{\mathcal{O}}\left(\left(\frac{1}{\varepsilon}\right)^{2\kappa-2}\right)$ | $\tilde{\mathcal{O}}\left(\left(\frac{1}{\varepsilon}\right)^{2\kappa-2}\theta+d\theta\right)$ | $\mathcal{O}(\varepsilon^{2\kappa})$ |
| Adversarial ($\nu=\mathcal{O}(\varepsilon)$) | [19] | $\tilde{\mathcal{O}}(d\theta)$ | $\tilde{\mathcal{O}}(d\theta)$ | N/A |
| Adversarial ($\nu=\mathcal{O}(\varepsilon)$) | **Ours** | $\tilde{\mathcal{O}}(1)$ | $\tilde{\mathcal{O}}(d\theta)$ | $\mathcal{O}(\varepsilon^2)$ |

humans find it easier to perform pairwise comparisons rather than providing direct labels, including content search [17], image retrieval [31], ranking [21], etc.

Despite many successful applications of comparison oracles, many fundamental questions remain. One of them is how to design *noise-tolerant*, *cost-efficient* algorithms that can approximate the unknown target hypothesis to arbitrary accuracy while having access to pairwise comparisons. On one hand, while there is theoretical analysis on the pairwise comparisons concerning the task of learning to rank [3, 22], estimating ordinal measurement models [28] and learning combinatorial functions [11], much remains unknown how to extend these results to more generic hypothesis classes. On the other hand, although we have seen great progress on using single or multiple oracles with the same form of interaction [9, 16], classification using both comparison and labeling queries remains an interesting open problem. Independently of our work, Kane et al. [23] concurrently analyzed a similar setting of learning to classify using both label and comparison queries. However, their algorithms work only in the noise-free setting.

**Our Contributions:** Our work addresses the aforementioned issues by presenting a new algorithm, Active Data Generation with Adversarial Comparisons (ADGAC), which learns a classifier with both noisy labeling and noisy comparison oracles.

- We analyze ADGAC under Tsybakov (TNC) [30] and adversarial noise conditions for the labeling oracle, along with the adversarial noise condition for the comparison oracle. Our general framework can augment any active learning algorithm by replacing the batch sampling in these algorithms with ADGAC. Figure 1 presents the work flow of our framework.

- We propose $A^2$-ADGAC algorithm, which can learn an arbitrary hypothesis class. The label complexity of the algorithm is as small as learning a threshold function under both TNC and adversarial noise condition, independently of the structure of the hypothesis class. The *total query complexity* improves over previous best-known results under TNC which can only access the labeling oracle.

- We derive Margin-ADGAC to learn the class of halfspaces. This algorithm has the same label and total query complexity as $A^2$-ADGAC, but is computationally efficient.

- We present lower bounds on total query complexity for any algorithm that can access both labeling and comparison oracles, and a noise tolerance lower bound for our algorithms. These lower bounds demonstrate that our analysis is nearly optimal.

An important quantity governing the performance of our algorithms is the adversarial noise level of comparisons: denote by $\mathsf{Tol}_{\text{comp}}(\varepsilon, \delta, \mathcal{A})$ the adversarial noise tolerance level of comparisons that guarantees an algorithm $\mathcal{A}$ to achieve an error of $\varepsilon$, with probability at least $1-\delta$. Table 1 compares our results with previous work in terms of label complexity, total query complexity, and $\mathsf{Tol}_{\text{comp}}$ for generic hypothesis class $\mathbb{C}$ with error $\varepsilon$. We see that our results significantly improve over prior

Table 2: Comparison of various methods for learning of halfspaces (Omitting $\log(1/\varepsilon)$ factors).

| Label Noise | Work | # Label | # Query | $\mathsf{Tol}_{\text{comp}}$ | Efficient? |
|---|---|---|---|---|---|
| Massart | [8] | $\tilde{\mathcal{O}}(d)$ | $\tilde{\mathcal{O}}(d)$ | N/A | No |
| Massart | [5] | $\text{poly}(d)$ | $\text{poly}(d)$ | N/A | Yes |
| Massart | **Ours** | $\tilde{\mathcal{O}}(1)$ | $\tilde{\mathcal{O}}(d)$ | $\mathcal{O}(\varepsilon^2)$ | Yes |
| Tsybakov ($\kappa$) | [19] | $\tilde{\mathcal{O}}((\frac{1}{\varepsilon})^{2\kappa-2}d\theta)$ | $\tilde{\mathcal{O}}((\frac{1}{\varepsilon})^{2\kappa-2}d\theta)$ | N/A | No |
| Tsybakov ($\kappa$) | **Ours** | $\tilde{\mathcal{O}}\left((\frac{1}{\varepsilon})^{2\kappa-2}\right)$ | $\tilde{\mathcal{O}}\left((\frac{1}{\varepsilon})^{2\kappa-2}+d\right)$ | $\mathcal{O}(\varepsilon^{2\kappa})$ | Yes |
| Adversarial ($\nu=\mathcal{O}(\varepsilon)$) | [34] | $\tilde{\mathcal{O}}(d)$ | $\tilde{\mathcal{O}}(d)$ | N/A | No |
| Adversarial ($\nu=\mathcal{O}(\varepsilon)$) | [6] | $\tilde{\mathcal{O}}(d^2)$ | $\tilde{\mathcal{O}}(d^2)$ | N/A | Yes |
| Adversarial ($\nu=\mathcal{O}(\varepsilon)$) | **Ours** | $\tilde{\mathcal{O}}(1)$ | $\tilde{\mathcal{O}}(d)$ | $\mathcal{O}(\varepsilon^2)$ | Yes |

work with the extra comparison oracle. Denote by $d$ the VC-dimension of $\mathbb{C}$ and $\theta$ the disagreement coefficient. We also compare the results in Table 2 for learning halfspaces under isotropic log-concave distributions. In both cases, our algorithms enjoy small label complexity that is independent of $\theta$ and $d$. This is helpful when labels are very expensive to obtain. Our algorithms also enjoy better total query complexity under both TNC and adversarial noise condition for efficiently learning halfspaces.

## 2 Preliminaries

**Notations:** We study the problem of learning a classifier $h : \mathcal{X} \to \mathcal{Y} = \{-1, 1\}$, where $\mathcal{X}$ and $\mathcal{Y}$ are the instance space and label space, respectively. Denote by $\mathcal{P}_{\mathcal{X}\mathcal{Y}}$ the distribution over $\mathcal{X} \times \mathcal{Y}$ and let $\mathcal{P}_{\mathcal{X}}$ be the marginal distribution over $\mathcal{X}$. A hypothesis class $\mathbb{C}$ is a set of functions $h : \mathcal{X} \to \mathcal{Y}$. For any function $h$, define the error of $h$ under distribution $D$ over $\mathcal{X} \times \mathcal{Y}$ as $\mathsf{err}_D(h) = \Pr_{(X,Y)\sim D}[h(X) \neq Y]$. Let $\mathsf{err}(h) = \mathsf{err}_{\mathcal{P}_{\mathcal{X}\mathcal{Y}}}(h)$. Suppose that $h^* \in \mathbb{C}$ satisfies $\mathsf{err}(h^*) = \inf_{h\in\mathbb{C}} \mathsf{err}(h)$. For simplicity, we assume that such an $h^*$ exists in class $\mathbb{C}$.

We apply the concept of disagreement coefficient from Hanneke [18] for generic hypothesis class in this paper. In particular, for any set $V \subseteq \mathbb{C}$, we denote by $\mathsf{DIS}(V) = \{x \in \mathcal{X} : \exists h_1, h_2 \in V, h_1(x) \neq h_2(x)\}$. The disagreement coefficient is defined as $\theta = \sup_{r>0} \frac{\Pr[\mathsf{DIS}(B(h^*,r))]}{r}$, where $B(h^*, r) = \{h \in \mathbb{C} : \Pr_{X\sim\mathcal{P}_{\mathcal{X}}}[h(X) \neq h^*(X)] \leq r\}$.

**Problem Setup:** We analyze two kinds of noise conditions for the labeling oracle, namely, adversarial noise condition and Tsybakov noise condition (TNC). We formally define them as follows.

**Condition 1** (Adversarial Noise Condition for Labeling Oracle). *Distribution $\mathcal{P}_{\mathcal{X}\mathcal{Y}}$ satisfies adversarial noise condition for labeling oracle with parameter $\nu \geq 0$, if $\nu = \Pr_{(X,Y)\sim\mathcal{P}_{\mathcal{X}\mathcal{Y}}}[Y \neq h^*(X)]$.*

**Condition 2** (Tsybakov Noise Condition for Labeling Oracle). *Distribution $\mathcal{P}_{\mathcal{X}\mathcal{Y}}$ satisfies Tsybakov noise condition for labeling oracle with parameters $\kappa \geq 1, \mu \geq 0$, if $\forall h : \mathcal{X} \to \{-1, 1\}, \mathsf{err}(h) - \mathsf{err}(h^*) \geq \mu \Pr_{X\sim\mathcal{P}_{\mathcal{X}}}[h(X) \neq h^*(X)]^{\kappa}$. Also, $h^*$ is the Bayes optimal classifier, i.e., $h^*(x) = \text{sign}(\eta(x) - 1/2)$.* [1] *where $\eta(x) = \Pr[Y = 1|X = x]$. The special case of $\kappa = 1$ is also called Massart noise condition.*

In the classic active learning scenario, the algorithm has access to an unlabeled pool drawn from $\mathcal{P}_{\mathcal{X}}$. The algorithm can then query the labeling oracle for any instance from the pool. The goal is to find an $h \in \mathbb{C}$ such that the error $\Pr[h(X) \neq h^*(X)] \leq \varepsilon^2$. The labeling oracle has access to the input $x \in \mathcal{X}$, and outputs $y \in \{-1, 1\}$ according to $\mathcal{P}_{\mathcal{X}\mathcal{Y}}$. In our setting, however, an extra comparison oracle is available. This oracle takes as input a pair of instances $(x, x') \in \mathcal{X} \times \mathcal{X}$, and returns a variable $Z(x, x') \in \{-1, 1\}$, where $Z(x, x') = 1$ indicates that $x$ is more likely to be positive, while $Z(x, x') = -1$ otherwise. In this paper, we discuss an adversarial noise condition for the comparison oracle. We discuss about dealing with TNC on the comparison oracle in appendix.

**Condition 3** (Adversarial Noise Condition for Comparison Oracle). *Distribution $\mathcal{P}_{\mathcal{X}\mathcal{X}\mathcal{Z}}$ satisfies adversarial noise with parameter $\nu' \geq 0$, if $\nu' = \Pr[Z(X, X')(h^*(X) - h^*(X')) < 0]$.*

Table 3: Summary of notations.

| Notation | Meaning | Notation | Meaning |
|---|---|---|---|
| $\mathbb{C}$ | Hypothesis class | $\kappa$ | Tsybakov noise level (labeling) |
| $X, \mathcal{X}$ | Instance & Instance space | $\nu$ | Adversarial noise level (labeling) |
| $Y, \mathcal{Y}$ | Label & Label space | $\nu'$ | Adversarial noise level (comparison) |
| $Z, \mathcal{Z}$ | Comparison & Comparison space | $\text{err}_D(h)$ | Error of $h$ on distribution $D$ |
| $d$ | VC dimension of $\mathbb{C}$ | $\text{SC}_{\text{label}}$ | Label complexity |
| $\theta$ | Disagreement coefficient | $\text{SC}_{\text{comp}}$ | Comparison complexity |
| $h^*$ | Optimal classifier in $\mathbb{C}$ | $\text{Tol}_{\text{label}}$ | Noise tolerance (labeling) |
| $g^*$ | Optimal scoring function | $\text{Tol}_{\text{comp}}$ | Noise tolerance (comparison) |

Note that we do not make any assumptions on the randomness of $Z$: $Z(X, X')$ can be either random or deterministic as long as the joint distribution $P_{\mathcal{X}\mathcal{X}\mathcal{Z}}$ satisfies Condition 3.

For an interactive learning algorithm $\mathcal{A}$, given error $\varepsilon$ and failure probability $\delta$, let $\text{SC}_{\text{comp}}(\varepsilon, \delta, \mathcal{A})$ and $\text{SC}_{\text{label}}(\varepsilon, \delta, \mathcal{A})$ be the comparison and label complexity, respectively. The query complexity of $\mathcal{A}$ is defined as the sum of label and comparison complexity. Similar to the definition of $\text{Tol}_{\text{comp}}(\varepsilon, \delta, \mathcal{A})$, define $\text{Tol}_{\text{label}}(\varepsilon, \delta, \mathcal{A})$ as the maximum $\nu$ such that algorithm $\mathcal{A}$ achieves an error of at most $\varepsilon$ with probability $1 - \delta$. As a summary, $\mathcal{A}$ learns an $h$ such that $\Pr[h(X) \neq h^*(X)] \leq \varepsilon$ with probability $1 - \delta$ using $\text{SC}_{\text{comp}}(\varepsilon, \delta, \mathcal{A})$ comparisons and $\text{SC}_{\text{label}}(\varepsilon, \delta, \mathcal{A})$ labels, if $\nu \leq \text{Tol}_{\text{label}}(\varepsilon, \delta, \mathcal{A})$ and $\nu' \leq \text{Tol}_{\text{comp}}(\varepsilon, \delta, \mathcal{A})$. We omit the parameters of $\text{SC}_{\text{comp}}, \text{SC}_{\text{label}}, \text{Tol}_{\text{comp}}, \text{Tol}_{\text{label}}$ if they are clear from the context. We use $\mathcal{O}(\cdot)$ to express sample complexity and noise tolerance, and $\tilde{O}(\cdot)$ to ignore the $\log(\cdot)$ terms. Table 3 summarizes the main notations throughout the paper.

## 3  Active Data Generation with Adversarial Comparisons (ADGAC)

The hardness of learning from pairwise comparisons follows from the error of comparison oracle: the comparisons are *noisy*, and can be *asymmetric* and *intransitive*, meaning that the human might give contradicting preferences like $x_1 \preccurlyeq x_2 \preccurlyeq x_1$ or $x_1 \preccurlyeq x_2 \preccurlyeq x_3 \preccurlyeq x_1$ (here $\preccurlyeq$ is some preference). This makes traditional methods, e.g., defining a function class $\{h : h(x) = Z(x, \hat{x}), \hat{x} \in \mathcal{X}\}$, fail, because such a class may have infinite VC dimension.

In this section, we propose a novel algorithm, ADGAC, to address this issue. Having access to both comparison and labeling oracles, ADGAC generates a labeled dataset by techniques inspired from group-based binary search. We show that ADGAC can be combined with any active learning procedure to obtain interactive algorithms that can utilize both labeling and comparison oracles. We provide theoretical guarantees for ADGAC.

### 3.1  Algorithm Description

To illustrate ADGAC, we start with a general active learning framework in Algorithm 1. Many active learning algorithms can be adapted to this framework, such as $A^2$ [7] and margin-based active algorithms [6, 5]. Here $U$ represents the querying space/disagreement region of the algorithm (i.e., we reject an instance $x$ if $x \notin U$), and $V$ represents a version space consisting of potential classifiers. For example, $A^2$ algorithm can be adapted to Algorithm 1 straightforwardly by keeping $U$ as the sample space and $V$ as the version space. More concretely, $A^2$ algorithm [7] for adversarial noise can be characterized by

$$U_0 = \mathcal{X}, \ V_0 = \mathbb{C}, \ f_V(U, V, W, i) = \{h : |W|\text{err}_W(h) \leq n_i\varepsilon_i\}, \ f_U(U, V, W, i) = \text{DIS}(V),$$

where $\varepsilon_i$ and $n_i$ are parameters of the $A^2$ algorithm, and $\text{DIS}(V) = \{x \in \mathcal{X} : \exists h_1, h_2 \in V, h_1(x) \neq h_2(x)\}$ is the disagreement region of $V$. Margin-based active learning [6] can also be fitted into Algorithm 1 by taking $V$ as the halfspace that (approximately) minimizes the hinge loss, and $U$ as the region within the margin of that halfspace.

To efficiently apply the comparison oracle, we propose to replace step 4 in Algorithm 1 with a subroutine, ADGAC, that has access to both comparison and labeling oracles. Subroutine 2 describes ADGAC. It takes as input a dataset $S$ and a sampling number $k$. ADGAC first runs Quicksort algorithm on $S$ using feedback from comparison oracle, which is of form $Z(x, x')$. Given that the comparison oracle $Z(\cdot, \cdot)$ might be asymmetric w.r.t. its two arguments, i.e., $Z(x, x')$ may not equal to $Z(x', x)$, for each pair $(x_i, x_j)$, we randomly choose $(x_i, x_j)$ or $(x_j, x_i)$ as the input to $Z(\cdot, \cdot)$. After Quicksort, the algorithm divides the data into multiple groups of size $\alpha m = \varepsilon|\tilde{S}|$, and does

---

**Algorithm 1** Active Learning Framework

---

**Input:** $\varepsilon, \delta$, a sequence of $n_i$, functions $f_U, f_V$.
1: Initialize $U \leftarrow U_0 \subseteq \mathcal{X}, V \leftarrow V_0 \subseteq \mathbb{C}$.
2: **for** $i = 1, 2, ..., \log(1/\varepsilon)$ **do**
3:      Sample unlabeled dataset $\tilde{S}$ of size $n_i$. Let $S \leftarrow \{x : x \in \tilde{S}, x \in U\}$.
4:      Request the labels of $x \in S$ and obtain $W \leftarrow \{(x_i, y_i) : x_i \in S\}$.
5:      Update $V \leftarrow f_V(U, V, W, i), U \leftarrow f_U(U, V, W, i)$.
**Output:** Any classifier $\hat{h} \in V$.

---

---

**Subroutine 2** Active Data Generation with Adversarial Comparison (ADGAC)

---

**Input:** Dataset $S$ with $|S| = m, n, \varepsilon, k$.
1: $\alpha \leftarrow \frac{\varepsilon n}{2m}$.
2: Define preference relation on $S$ according to $Z$. Run Quicksort on $S$ to rank elements in an increasing order. Obtain a sorted list $S = (x_1, x_2, ..., x_m)$.
3: Divide $S$ into groups of size $\alpha m$: $S_i = \{x_{(i-1)\alpha m+1}, ..., x_{i\alpha m}\}, i = 1, 2, ..., 1/\alpha$ .
4: $t_{\min} \leftarrow 1, t_{\max} \leftarrow 1/\alpha$.
5: **while** $t_{\min} < t_{\max}$ **do**                                            ▷ Do binary search
6:      $t = (t_{\min} + t_{\max})/2$.
7:      Sample $k$ points uniformly without replacement from $S_t$ and obtain the labels $Y = \{y_1, ..., y_k\}$.
8:      **If** $\sum_{i=1}^{k} y_i \geq 0$, **then** $t_{\max} = t$; **else** $t_{\min} = t + 1$.
9: For $t' > t$ and $x_i \in S_{t'}$, let $\hat{y}_i \leftarrow 1$.
10: For $t' < t$ and $x_i \in S_{t'}$, let $\hat{y}_i \leftarrow -1$.
11: For $x_i \in S_t$, let $\hat{y}_i$ be the majority of labeled points in $S_t$.
**Output:** Predicted labels $\hat{y}_1, \hat{y}_2, ..., \hat{y}_m$.

---

group-based binary search by sampling $k$ labels from each group and determining the label of each group by majority vote.

For active learning algorithm $\mathcal{A}$, let $\mathcal{A}$-ADGAC be the algorithm of replacing step 4 with ADGAC using parameters $(S_i, n_i, \varepsilon_i, k_i)$, where $\varepsilon_i, k_i$ are chosen as additional parameters of the algorithm. We establish results for specific $\mathcal{A}$: $A^2$ and margin-based active learning in Sections 4 and 5, respectively.

### 3.2 Theoretical Analysis of ADGAC

Before we combine ADGAC with active learning algorithms, we provide theoretical results for ADGAC. By the algorithmic procedure, ADGAC reduces the problem of labeling the whole dataset $S$ to binary searching a threshold on the sorted list $S$. One can show that the conflicting instances cannot be too many within each group $S_i$, and thus binary search performs well in our algorithm. We also use results in [3] to give an error estimate of Quicksort. We have the following result based on the above arguments.

**Theorem 4.** *Suppose that Conditions 2 and 3 hold for $\kappa \geq 1, \nu' \geq 0$, and $n = \Omega\left(\left(\frac{1}{\varepsilon}\right)^{2\kappa-1} \log(1/\delta)\right)$. Assume a set $\tilde{S}$ with $|\tilde{S}| = n$ is sampled i.i.d. from $\mathcal{P}_\mathcal{X}$ and $S \subseteq \tilde{S}$ is an arbitrary subset of $\tilde{S}$ with $|S| = m$. There exist absolute constants $C_1, C_2, C_3$ such that if we run Subroutine 2 with $\varepsilon < C_1, \nu' \leq C_2\varepsilon^{2\kappa}\delta, k = k^{(1)}(\varepsilon, \delta) := C_3 \log\left(\frac{\log(1/\varepsilon)}{\delta}\right)\left(\frac{1}{\varepsilon}\right)^{2\kappa-2}$, it will output a labeling of $S$ such that $|\{x_i \in S : \hat{y}_i \neq h^*(x_i)\}| \leq \varepsilon n$, with probability at least $1 - \delta$. The expected number of comparisons required is $\mathcal{O}(m \log m)$, and the number of sample-label pairs required is $\mathsf{SC}_{\text{label}}(\varepsilon, \delta) = \tilde{\mathcal{O}}\left(\log\left(\frac{m}{\varepsilon n}\right) \log(1/\delta) \left(\frac{1}{\varepsilon}\right)^{2\kappa-2}\right)$.*

Similarly, we analyze ADGAC under adversarial noise condition w.r.t. labeling oracle with $\nu = \mathcal{O}(\varepsilon)$.

**Theorem 5.** *Suppose that Conditions 1 and 3 hold for $\nu, \nu' \geq 0$, and $n = \Omega\left(\frac{1}{\varepsilon}\log(1/\delta)\right)$. Assume a set $\tilde{S}$ with $|\tilde{S}| = n$ is sampled i.i.d. from $\mathcal{P}_\mathcal{X}$ and $S \subseteq \tilde{S}$ is an arbitrary subset of $\tilde{S}$ with $|S| = m$. There exist absolute constants $C_1, C_2, C_3, C_4$ such that if we run Subroutine 2 with $\varepsilon < C_1, \nu' \leq C_2\varepsilon^2\delta, k = k^{(2)}(\varepsilon, \delta) := C_3 \log\left(\frac{\log(1/\varepsilon)}{\delta}\right)$, and $\nu \leq C_4\varepsilon$, it will output a labeling*

of $S$ such that $|\{x_i \in S : \hat{y}_i \neq h^*(x_i)\}| \leq \varepsilon n$, with probability at least $1 - \delta$. The expected number of comparisons required is $\mathcal{O}(m \log m)$, and the number of sample-label pairs required is $\mathsf{SC}_{\text{label}}(\varepsilon, \delta) = \mathcal{O}\left(\log\left(\frac{m}{\varepsilon n}\right) \log\left(\frac{\log(1/\varepsilon)}{\delta}\right)\right)$.

**Proof Sketch.** We call a pair $(x_i, x_j)$ an *inverse pair* if $Z(x_i, x_j) = -1, h^*(x_i) = 1, h^*(x_j) = -1$, and an *anti-sort pair* if $h^*(x_i) = 1, h^*(x_j) = -1$, and $i < j$. We show that the expectation of inverse pairs is $n(n-1)\varepsilon^*$. By the results in [3] the numbers of inverse pairs and anti-sort pairs have the same expectation, and the actual number of anti-sort pairs can be bounded by Markov's inequality. Then we show that the majority label of each group must be all -1 starting from beginning the list, and changes to all 1 at some point of the list. With a careful choice of $k$, we may obtain the true majority with $k$ labels under Tsybakov noise; we will thus end up in the turning point of the list. The error is then bounded by the size of groups. See appendix for the complete proof.

Theorems 4 and 5 show that ADGAC gives a labeling of dataset with arbitrary small error using label complexity *independent* of the data size. Moreover, ADGAC is computationally efficient since it only involves binary search. These nice properties of ADGAC lead to improved query complexity when we combine ADGAC with other active learning algorithms.

# 4   $A^2$-ADGAC: Learning of Generic Hypothesis Class

In this section, we combine ADGAC with $A^2$ algorithm to learn a generic hypothesis class. We use the framework in Algorithm 1: let $A^2$-ADGAC be the algorithm that replaces step 4 in Algorithm 1 with ADGAC of parameters $(S, n_i, \varepsilon_i, k_i)$, where $n_i, \varepsilon_i, k_i$ are parameters to be specified later. Under TNC, we have the following result.

**Theorem 6.** *Suppose that Conditions 2 and 3 hold, and $h^*(x) = sign(\eta(x) - 1/2)$. There exist global constants $C_1, C_2$ such that if we run $A^2$-ADGAC with $\varepsilon < C_1, \delta, \nu' \leq \mathsf{Tol}_{\text{comp}}(\varepsilon, \delta) = C_2\varepsilon^{2\kappa}\delta$,*
$\varepsilon_i = 2^{-(i+2)}, n_i = \Omega\left(\frac{1}{\varepsilon_i}(d\log(1/\varepsilon)) + \left(\frac{1}{\varepsilon_i}\right)^{2\kappa-1}\log(1/\delta)\right), k_i = k^{(1)}\left(\varepsilon_i, \frac{\delta}{4\log(1/\varepsilon)}\right)$ *with* $k^{(1)}$
*specified in Theorem 4, with probability at least $1 - \delta$, the algorithm will return a classifier $\hat{h}$ with $\Pr[\hat{h}(X) \neq h^*(X)] \leq \varepsilon$ with comparison and label complexity*

$$\mathbb{E}[\mathsf{SC}_{\text{comp}}] = \tilde{\mathcal{O}}\left(\theta\log^2\left(\frac{1}{\varepsilon}\right)\log(d\theta)\left(\left(d\log\left(\frac{1}{\varepsilon}\right)\right) + \left(\frac{1}{\varepsilon}\right)^{2\kappa-2}\log(1/\delta)\right)\right),$$

$$\mathsf{SC}_{\text{label}} = \tilde{\mathcal{O}}\left(\log\left(\frac{1}{\varepsilon}\right)\log\left(\min\left\{\frac{1}{\varepsilon}, \theta\right\}\right)\log(1/\delta)\left(\frac{1}{\varepsilon}\right)^{2\kappa-2}\right).$$

*The dependence on $\log^2(1/\varepsilon)$ in $\mathsf{SC}_{\text{comp}}$ can be reduced to $\log(1/\varepsilon)$ under Massart noise.*

We can prove a similar result for adversarial noise condition.

**Theorem 7.** *Suppose that Conditions 1 and 3 hold. There exist global constants $C_1, C_2, C_3$ such that if we run $A^2$-ADGAC with $\varepsilon < C_1, \delta, \nu' \leq \mathsf{Tol}_{\text{comp}}(\varepsilon, \delta) = C_2\varepsilon^2\delta, \nu \leq \mathsf{Tol}_{\text{label}}(\varepsilon, \delta) = C_3\varepsilon, \varepsilon_i = 2^{-(i+2)}, n_i = \tilde{\Omega}\left(\frac{1}{\varepsilon_i}d\log\left(\frac{1}{\varepsilon_i}\right)\log(1/\delta)\right), k_i = k^{(2)}\left(\varepsilon_i, \frac{\delta}{4\log(1/\varepsilon)}\right)$ with $k^{(2)}$ specified in Theorem 5, with probability at least $1 - \delta$, the algorithm will return a classifier $\hat{h}$ with $\Pr[\hat{h}(X) \neq h^*(X)] \leq \varepsilon$ with comparison and label complexity*

$$\mathbb{E}[\mathsf{SC}_{\text{comp}}] = \tilde{\mathcal{O}}\left(\theta d\log(\theta d)\log\left(\frac{1}{\varepsilon_i}\right)\log(1/\delta)\right),$$

$$\mathsf{SC}_{\text{label}} = \tilde{\mathcal{O}}\left(\log\left(\frac{1}{\varepsilon}\right)\log\left(\min\left\{\frac{1}{\varepsilon}, \theta\right\}\right)\log(1/\delta)\right).$$

Proof of Theorems 6 and 7 uses Theorem 4 and Theorem 5 with standard manipulations in VC theory. Theorems 6 and 7 show that having access to even a biased comparison function can reduce the problem of learning a classifier in high-dimensional space to that of learning a threshold classifier in one-dimensional space as the label complexity matches that of actively learning a threshold classifier. Given the fact that comparisons are usually easier to obtain, $A^2$-ADGAC will save a lot in practice due to its small label complexity. More importantly, we improve the total query complexity under TNC by separating the dependence on $d$ and $\varepsilon$; The query complexity is now the sum of the two terms instead of the product of them. This observation shows the power of pairwise comparisons for learning classifiers. Such small label/query complexity is impossible without access to a comparison

oracle, since query complexity with only labeling oracle is at least $\Omega\left(d\left(\frac{1}{\varepsilon}\right)^{2\kappa-2}\right)$ and $\Omega\left(d\log\left(\frac{1}{\varepsilon}\right)\right)$ under TNC and adversarial noise conditions, respectively [19]. Our results also matches the lower bound of learning with labeling and comparison oracles up to log factors (see Section 6).

We note that Theorems 6 and 7 require rather small $\mathsf{Tol}_{\text{comp}}$, equal to $\mathcal{O}(\varepsilon^{2\kappa}\delta)$ and $\mathcal{O}(\varepsilon^2\delta)$, respectively. We will show in Section 6.3 that it is necessary to require $\mathsf{Tol}_{\text{comp}} = \mathcal{O}(\varepsilon^2)$ in order to obtain a classifier of error $\varepsilon$, if we restrict the use of labeling oracle to only learning a threshold function. Such restriction is able to reach the near-optimal label complexity as specified in Theorems 6 and 7.

## 5 Margin-ADGAC: Learning of Halfspaces

In this section, we combine ADGAC with margin-based active learning [6] to efficiently learn the class of halfspaces. Before proceeding, we first mention a naive idea of utilizing comparisons: we can i.i.d. sample pairs $(x_1, x_2)$ from $\mathcal{P}_{\mathcal{X}} \times \mathcal{P}_{\mathcal{X}}$, and use $Z(x_1, x_2)$ as the label of $x_1 - x_2$, where $Z$ is the feedback from comparison oracle. However, this method cannot work well in our setting without additional assumption on the noise condition for the labeling $Z(x_1, x_2)$.

Before proceeding, we assume that $\mathcal{P}_{\mathcal{X}}$ is isotropic log-concave on $\mathbb{R}^d$; i.e., $\mathcal{P}_{\mathcal{X}}$ has mean 0, covariance $I$ and the logarithm of its density function is a concave function [5, 6]. The hypothesis class of halfspaces can be represented as $\mathbb{C} = \{h : h(x) = \text{sign}(w \cdot x), w \in \mathbb{R}^d\}$. Denote by $h^*(x) = \text{sign}(w^* \cdot x)$ for some $w^* \in \mathbb{R}^d$. Define $l_\tau(w, x, y) = \max\left(1 - y(w \cdot x)/\tau, 0\right)$ and $l_\tau(w, W) = \frac{1}{|W|}\sum_{(x,y)\in W} l_\tau(w, x, y)$ as the hinge loss. The expected hinge loss of $w$ is $L_\tau(w, D) = \mathbb{E}_{x\sim D}[l_\tau(w, x, \text{sign}(w^* \cdot x))]$.

Margin-based active learning [6] is a concrete example of Algorithm 1 by taking $V$ as (a singleton set of) the hinge loss minimizer, while taking $U$ as the margin region around that minimizer. More concretely, take $U_0 = \mathcal{X}$ and $V_0 = \{w_0\}$ for some $w_0$ such that $\theta(w_0, w^*) \leq \pi/2$. The algorithm works with constants $M \geq 2, \kappa < 1/2$ and a set of parameters $r_i, \tau_i, b_i, z_i$ that equal to $\Theta(M^{-i})$ (see proof in Appendix for formal definition of these parameters). $V$ always contains a single hypothesis. Suppose $V = \{w_{i-1}\}$ in iteration $i - 1$. Let $v_i$ satisfies $l_{\tau_i}(v_i, W) \leq \min_{v:\|v-w_{i-1}\|_2 \leq r_i, \|v\|_2 \leq 1} l_{\tau_i}(v, W) + \kappa/8$, where $w_i$ is the content of $V$ in iteration $i$. We also have $f_V(V, W, i) = \{w_i\} = \left\{\frac{v_i}{\|v_i\|_2}\right\}$ and $f_U(U, V, W, i) = \{x : |w_i \cdot x| \leq b_i\}$.

Let Margin-ADGAC be the algorithm obtained by replacing the sampling step in margin-based active learning with ADGAC using parameters $(S, n_i, \varepsilon_i, k_i)$, where $n_i, \varepsilon_i, k_i$ are additional parameters to be specified later. We have the following results under TNC and adversarial noise conditions, respectively.

**Theorem 8.** *Suppose that Conditions 2 and 3 hold, and $h^*(x) = sign(w^* \cdot x) = sign(\eta(x) - 1/2)$. There are settings of $M, \kappa, r_i, \tau_i, b_i, \varepsilon_i, k_i,$ and constants $C_1, C_2$ such that for all $\varepsilon \leq C_1, \nu' \leq \mathsf{Tol}_{\text{comp}}(\varepsilon, \delta) = C_2\varepsilon^{2\kappa}\delta$, if we run Margin-ADGAC with $w_0$ such that $\theta(w_0, w^*) \leq \pi/2$, and $n_i = \tilde{\mathcal{O}}\left(\frac{1}{\varepsilon_i}d\log^3(dk/\delta) + \left(\frac{1}{\varepsilon}\right)^{2\kappa-1}\log(1/\delta)\right)$, it finds $\hat{w}$ such that $\Pr[sign(\hat{w} \cdot X) \neq sign(w^* \cdot X)] \leq \varepsilon$ with probability at least $1 - \delta$. The comparison and label complexity are*

$$\mathbb{E}[\mathsf{SC}_{\text{comp}}] = \tilde{\mathcal{O}}\left(\log^2(1/\varepsilon)\left(d\log^4(d/\delta) + \left(\frac{1}{\varepsilon}\right)^{2\kappa-2}\log(1/\delta)\right)\right),$$

$$\mathsf{SC}_{\text{label}} = \tilde{\mathcal{O}}\left(\log(1/\varepsilon)\log(1/\delta)\left(\frac{1}{\varepsilon}\right)^{2\kappa-2}\right).$$

*The dependence on $\log^2(1/\varepsilon)$ in $\mathsf{SC}_{\text{comp}}$ can be reduced to $\log(1/\varepsilon)$ under Massart noise.*

**Theorem 9.** *Suppose that Conditions 1 and 3 hold. There are settings of $M, \kappa, r_i, \tau_i, b_i, \varepsilon_i, k_i,$ and constants $C_1, C_2, C_3$ such that for all $\varepsilon \leq C_1, \nu' \leq \mathsf{Tol}_{\text{comp}}(\varepsilon, \delta) = C_2\varepsilon^{2\kappa}\delta, \nu \leq \mathsf{Tol}_{\text{comp}}(\varepsilon, \delta) = C_3\varepsilon$, if we run Margin-ADGAC with $n_i = \tilde{\mathcal{O}}\left(\frac{1}{\varepsilon_i}d\log^3(dk/\delta)\right)$ and $w_0$ such that $\theta(w_0, w^*) \leq \pi/2$, it finds $\hat{w}$ such that $\Pr[sign(\hat{w} \cdot X) \neq sign(w^* \cdot X)] \leq \varepsilon$ with probability at least $1 - \delta$. The comparison and label complexity are*

$$\mathbb{E}[\mathsf{SC}_{\text{comp}}] = \tilde{\mathcal{O}}\left(\log(1/\varepsilon)\left(d\log^4(d/\delta)\right)\right), \quad \mathsf{SC}_{\text{label}} = \tilde{\mathcal{O}}\left(\log(1/\varepsilon)\log(1/\delta)\right).$$

The proofs of Theorems 8 and 9 are different from the conventional analysis of margin-based active learning in two aspects: a) Since we use labels generated by ADGAC, which is not independently

sampled from the distribution $\mathcal{P}_{\mathcal{X}\mathcal{Y}}$, we require new techniques that can deal with adaptive noises; b) We improve the results of [6] over the dependence of $d$ by new Rademacher analysis.

Theorems 8 and 9 enjoy better label and query complexity than previous results (see Table 2). We mention that while Yan and Zhang [32] proposed a perceptron-like algorithm with label complexity as small as $\tilde{\mathcal{O}}(d \log(1/\varepsilon))$ under Massart and adversarial noise conditions, their algorithm works only under uniform distributions over the instance space. In contrast, our algorithm Margin-ADGAC works under broad log-concave distributions. The label and total query complexity of Margin-ADGAC improves over that of traditional active learning. The lower bounds in Section 6 show the optimality of our complexity.

## 6 Lower Bounds

In this section, we give lower bounds on learning using labeling and pairwise comparison. In Section 6.1, we give a lower bound on the optimal label complexity $\mathsf{SC}_{\text{label}}$. In Section 6.2 we use this result to give a lower bound on the total query complexity, i.e., the sum of comparison and label complexity. Our two methods match these lower bounds up to log factors. In Section 6.3, we additionally give an information-theoretic bound on $\mathsf{Tol}_{\text{comp}}$, which matches our algorithms in the case of Massart and adversarial noise.

Following from [19, 20], we assume that there is an underlying score function $g^*$ such that $h^*(x) = \text{sign}(g^*(x))$. Note that $g^*$ does not necessarily have relation with $\eta(x)$; We only require that $g^*(x)$ represents how likely a given $x$ is positive. For instance, in digit recognition, $g^*(x)$ represents how an image looks like a 7 (or 9); In the clinical setting, $g^*(x)$ measures the health condition of a patient. Suppose that the distribution of $g^*(X)$ is continuous, i.e., the probability density function exists and for every $t \in \mathbb{R}$, $\Pr[g^*(X) = t] = 0$.

### 6.1 Lower Bound on Label Complexity

The definition of $g^*$ naturally induces a comparison oracle $Z$ with $Z(x, x') = \text{sign}(g^*(x) - g^*(x'))$. We note that this oracle is invariant to shifting w.r.t. $g^*$, i.e., $g^*$ and $g^* + t$ lead to the same comparison oracle. As a result, we cannot distinguish $g^*$ from $g^* + t$ without labels. In other words, pairwise comparisons do not help in improving label complexity when we are learning a threshold function on $\mathbb{R}$, where all instances are in the natural order. So the label complexity of any algorithm is lower bounded by that of learning a threshold classifier, and we formally prove this in the following theorem.

**Theorem 10.** *For any algorithm $\mathcal{A}$ that can access both labeling and comparison oracles, sufficiently small $\varepsilon, \delta$, and any score function $g$ that takes at least two values on $\mathcal{X}$, there exists a distribution $P_{\mathcal{X}\mathcal{Y}}$ satisfying Condition 2 such that the optimal function is in the form of $h^*(x) = \text{sign}(g(x) + t)$ for some $t \in \mathbb{R}$ and*

$$\mathsf{SC}_{\text{label}}(\varepsilon, \delta, \mathcal{A}) = \Omega\left((1/\varepsilon)^{2\kappa-2} \log(1/\delta)\right). \tag{1}$$

*If $\mathcal{P}_{\mathcal{X}\mathcal{Y}}$ satisfies Condition 1 with $\nu = O(\varepsilon)$, $\mathsf{SC}_{\text{label}}$ satisfies (1) with $\kappa = 1$.*

The lower bound in Theorem 10 matches the label complexity of $\mathrm{A}^2$-ADGAC and Margin-ADGAC up to a log factor. So our algorithm is near-optimal.

### 6.2 Lower Bound on Total Query Complexity

We use Theorem 10 to give lower bounds on the total query complexity of any algorithm which can access both comparison and labeling oracles.

**Theorem 11.** *For any algorithm $\mathcal{A}$ that can access both labeling and comparison oracles, and sufficiently small $\varepsilon, \delta$, there exists a distribution $P_{\mathcal{X}\mathcal{Y}}$ satisfying Condition 2, such that*

$$\mathsf{SC}_{\text{comp}}(\varepsilon, \delta, \mathcal{A}) + \mathsf{SC}_{\text{label}}(\varepsilon, \delta, \mathcal{A}) = \Omega\left((1/\varepsilon)^{2\kappa-2} \log(1/\delta) + d \log(1/\varepsilon)\right). \tag{2}$$

*If $\mathcal{P}_{\mathcal{X}\mathcal{Y}}$ satisfies Condition 1 with $\nu = O(\varepsilon)$, $\mathsf{SC}_{\text{comp}} + \mathsf{SC}_{\text{label}}$ satisfies (2) with $\kappa = 1$.*

The first term of (2) follows from Theorem 10, whereas the second term follows from transforming a lower bound of active learning with access to only the labeling oracle. The lower bounds in Theorem 11 match the performance of $\mathrm{A}^2$-ADGAC and Margin-ADGAC up to log factors.

### 6.3 Adversarial Noise Tolerance of Comparisons

Note that label queries are typically expensive in practice. Thus it is natural to ask the following question: what is the minimal requirement on $\nu'$, given that we are only allowed to have access to minimal label complexity as in Theorem 10? We study this problem in this section. More concretely,

we study the requirement on $\nu'$ when we learn a threshold function using labels. Suppose that the comparison oracle gives feedback using a scoring function $\hat{g}$, i.e., $Z(x, x') = \text{sign}(\hat{g}(x) - \hat{g}(x'))$, and has error $\nu'$. We give a sharp minimax bound on the risk of the optimal classifier in the form of $h(x) = \text{sign}(\hat{g}(x) - t)$ for some $t \in \mathbb{R}$ below.

**Theorem 12.** *Suppose that* $\min\{\Pr[h^*(X) = 1], \Pr[h^*(X) = -1]\} \geq \sqrt{\nu'}$ *and both* $\hat{g}(X)$ *and* $g^*(X)$ *have probability density functions. If* $\hat{g}(X)$ *induces an oracle with error* $\nu'$, *then we have* $\min_t \max_{\hat{g}, g^*} \Pr[sign(\hat{g}(X) - t) \neq h^*(X)] = \sqrt{\nu'}$.

The proof is technical and omitted. By Theorem 12, we see that the condition of $\nu' = \varepsilon^2$ is necessary if labels from $g^*$ are only used to learn a threshold on $\hat{g}$. This matches our choice of $\nu'$ under Massart and adversarial noise conditions for labeling oracle (up to a factor of $\delta$).

# 7    Conclusion

We presented a general algorithmic framework, ADGAC, for learning with both comparison and labeling oracles. We proposed two variants of the base algorithm, $A^2$-ADGAC and Margin-ADGAC, to facilitate low query complexity under Tsybakov and adversarial noise conditions. The performance of our algorithms matches lower bounds for learning with both oracles. Our analysis is relevant to a wide range of practical applications where it is easier, less expensive, and/or less risky to obtain pairwise comparisons than labels.

There are multiple directions for future works. One improvement over our work is to show complexity bounds for excess risk $\text{err}(h) - \text{err}(h^*)$ instead of $\Pr[h \neq h^*]$. Also, our bound on comparison complexity is in expectation due to limits of quicksort; deriving concentration inequalities on the comparison complexity would be helpful. Also, an adaptive algorithm that applies to different levels of noise w.r.t. labels and comparisons would be interesting; i.e., use labels when comparisons are noisy and use comparisons when labels are noisy. Other directions include using comparisons (or more broadly, rankings) for other ML tasks like regression or matrix completion.

### Acknowledgments

This research is supported in part by AFRL grant FA8750-17-2-0212. We thank Chicheng Zhang for insightful ideas on improving results in [6] using Rademacher complexity.

## Footnotes

[1] The assumption that $h^*$ is Bayes optimal classifier can be relaxed if the approximation error of $h^*$ can be quantified under assumptions on the decision boundary (c.f. [15]).

[2] Note that we use the disagreement $\Pr[h(X) \neq h^*(X)]$ instead of the excess error $\mathsf{err}(h) - \mathsf{err}(h^*)$ in some of the other literatures. The two conditions can be linked by assuming a two-sided version of Tsybakov noise (see e.g., Audibert 2004).

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
