[Supplementary Material]

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

# A  Our Techniques

**Intransitivity:** The main challenge of learning with pairwise comparisons is that the comparisons might be *asymmetric* or *intransitive*. If we construct a classifier $h(x)$ by simply comparing $x$ with a fixed instance $\hat{x}$ by comparison oracle, then the concept class of classifiers $\{h : h(x) = Z(x, \hat{x}), \hat{x} \in \mathcal{X}\}$ will have infinite VC dimension, so the complexity will be as high as infinite if we apply the traditional tools of VC theory. To resolve the issue, we conduct a group-based binary search in ADGAC. The intuition is that by dividing the dataset into several ranked groups $S_1, S_2, ...,$ the majority of labels in each group can be stably decided if we sample enough examples from that group. Therefore, we are able to reduce the original problem in the high-dimensional space to the problem of learning a "threshold" function in one-dimension space. Then some straightforward approaches such as binary search learns the thresholding function.

**Combining with Active Learning Algorithms:** If the labels follow Tsybakov noise (i.e., Condition 2), the most straightforward method to combine ADGAC with existing algorithms is to combine ADGAC with an algorithm that uses the label oracle only and works under TNC. However, we cannot save query complexity if we follow this method. To see this, notice that in each round we need roughly $n_i = \tilde{\mathcal{O}}\left( d\theta \left( \frac{1}{\varepsilon_i} \right)^{2\kappa-1} \right)$ samples and $m_i = \tilde{\mathcal{O}}\left( d\theta \left( \frac{1}{\varepsilon_i} \right)^{2\kappa-2} \right)$ labels; if we use ADGAC, we can obtain a labeling of $n_i$ samples with at most $\varepsilon_i n_i \approx m_i$ errors with low label complexity. Suppose $N$ is the set of labels that ADGAC makes error on. However, since the outside active learning algorithm works under TNC, we will need to query labels in $N$ to make sure that the ADGAC labels follow TNC. That means our label complexity is still $m_i$, the same as the original algorithm. To avoid this problem, we combine ADGAC with algorithms under adversarial noise in all cases including TNC. This eliminates the need to query additional labels, and also reduces the query complexity.

**Handling Independence:** We mostly follow previous works on combining ADGAC with existing algorithms. However, since we now obtain labels from ADGAC instead of $\mathcal{P}_{\mathcal{X}\mathcal{Y}}$, the labels are not independently sampled, and we need to adapt the proof to our case. We use different methods for $A^2$-ADGAC and Margin-ADGAC: For the former, we use results from PAC learning to bound the error on all $n_i$ samples; for the latter, we decompose the error of any classifier $h$ on labels generated by ADGAC into two parts: The first part is caused by the error of ADGAC itself, and second is by $h$ on truthful labels. Using the above techniques enables us to circumvent the independence problem.

**Lower Bounds:** It is typically hard to provide a unified lower bound for multi-query learning framework, as several quantities are simultaneously involved in the analysis, e.g., the comparison complexity, the label complexity, the noise tolerance, etc. So traditional proof techniques for active learning, e.g., Le Cam's and Fano's bounds [15, 19], cannot be trivially applied to our setting. Instead, we prove lower bounds on one quantity by allowing arbitrary budgets of other quantities. Another non-trivial technique is in the proof of minimax bound for the adversarial noise level of comparison oracle (see Theorem 12): In the proof of upper bound, we divide the integral region w.r.t. the expectation into $n$ segments, each of size $1/n$, and the expectation is thus the limit when $n \to \infty$. We upper bound the discrete approximation of the integral by a careful calibration of noise on each segment for a fixed $n$, and then let $n \to \infty$. The proof then leads to a general inequality (Lemma 21), and it might be of independent interest.

# B  Additional Related Work

It is well known that people are better at comparison than labeling [29, 28]. It has been widely used to tackle problems in classification [26], clustering [24] and ranking [2, 17]. Balcan et al. [11] studied using pairwise comparisons to learn submodular functions on sets. Another related problem is bipartite ranking [1], which exactly does the opposite of our problem: Given a group of binary labels, learn a ranking function that rank positive samples higher than negative ones.

Interactive learning has wide application in the field of computer vision and natural language processing (see e.g., [31]). There are also abundant literatures on interactive ways to improve unsupervised and semi-supervised learning [24]. However, there lacks a general statistical analysis of interactive learning for traditional classification tasks. Balcan and Hanneke [9] analyze class conditional queries

(CCQ), where the user gives counterexamples to a given classification. Beygelzimer et al. [13] used a similar idea using search queries. However, their interactions requires a oracle that is usually stronger than the traditional labelers (i.e., we can simulate traditional active learning using such oracles), and is generally hard to deploy in practice. There turns out to be little general analysis on using a "weaker" interaction between human and computer. Balcan and Hanneke[9] studied an abstract query based notions from exact learning, but their analysis cannot handle queries that gives relation between samples (as comparisons do). Our work fits in this blank.

We compare our work to traditional label-based active learning [19], which has drawn a lot of attention in the society in recent years. Disagreement-based active learning has been shown to reach a near-optimal rate on classification problems [18]. Another line of research is margin-based active learning [5], which aims at computational efficiency of learning halfspaces, under the large-margin assumption.

## C   Learning under TNC for Comparisons

In this section we justify our choice of analyzing adversarial noise model for the comparison oracle. In fact, any algorithm using adversarial comparisons can be transformed into an algorithm using TNC comparisons, by treating learning comparison functions as a separate learning problem. Let $\mathbb{C}'$ be a hypothesis class consisting of comparison functions $f : \mathcal{X} \times \mathcal{X} \to \{-1, 1\}$. Suppose the optimal comparison function is $f^*(x, x') = \text{sign}(g^*(x) - g^*(x'))$, and Tsybakov noise condition holds for $((X, X'), Z)$ with some constant $\mu', \kappa'$; i.e., for any $f \in \mathbb{C}'$ we have

$$\Pr[f(X, X') \neq Z] - \Pr[f^*(X, X') \neq Z] \geq \mu' \Pr[f(X, X') \neq f^*(X, X')]^{\kappa'}.$$

Also suppose $f^*(x, x') = \text{sign}(\Pr[Z = 1 | X = x, X' = x'] - 1/2)$. Assume $\mathbb{C}'$ has VC-dimension $d'$ and disagreement coefficient $\theta'$, standard active learning requires $\Phi(\nu') = \tilde{\mathcal{O}}\left(\theta'\left(\frac{1}{\nu'}\right)^{2\kappa'-2}(d'\log(\theta') + \log(1/\delta))\log\left(\frac{1}{\nu'}\right)\right)$ samples to learn a comparison function of error $\nu'$ with probability $1 - \delta$. So an algorithm $\mathcal{A}$ for adversarial noise on comparisons can be automatically transformed into an algorithm $\mathcal{A}'$ for TNC on comparisons with $\mathsf{SC}_{\text{label}}(\mathcal{A}') = \mathsf{SC}_{\text{label}}(\mathcal{A})$ and $\mathsf{SC}_{\text{comp}}(\mathcal{A}) = \Phi(\mathsf{Tol}_{\text{comp}}(\mathcal{A}))$. So we only analyze adversarial noise for comparison in other parts of this paper.

## D   Proof of Theorem 4

*Proof.* We only prove the theorem for $\kappa > 1$, the case of $\kappa = 1$ holds with a similar proof. An equivalent condition (see e.g., page 341, [14]) for Condition 2 under $\kappa > 1$ is that there exists constant $\tilde{\mu} > 0$ such that for all $t > 0$ we have

$$\Pr(|\eta(x) - 1/2| < t) \leq \tilde{\mu} t^{1/(\kappa-1)}. \tag{3}$$

We use (3) instead of Condition 2 through out the proof.

To bound the error in labeling by ADGAC, we first bound the number of incorrectly sorted pairs due to noise/bias of the comparison oracle. We call $(x_i, x_j)$ an *inverse pair* if $h^*(x_i) = 1, h^*(x_j) = -1, x_i \preccurlyeq x_j$ (the partial order is decided by randomly querying $Z(x_i, x_j)$ or $Z(x_j, x_i)$). Also, we call $(x_i, x_j)$ an *anti-sort pair* if $h^*(x_i) = 1, h^*(x_j) = -1, i < j$ (after sorting by Quicksort). Let $T$ be the set of all anti-sort pairs, $T'$ be the set of all inverse pairs in $S$, and $\tilde{T}'$ be the set of all inverse pairs in $\tilde{S}$. We first bound $|T|$ using $|T'|$. Let $s$ be the random bits supplied for Quicksort in its process, by Theorem 3 in [3] we have

$$\mathbb{E}_s[|T|] = |T'|.$$

Notice that sampling a pair of $(X, X')$ is equivalent to sample a set $\tilde{S}$ of $n$ points and then uniformly pick two different points in it. Also, number of inverse pairs in $S$ is less than that in $\tilde{S}$. So we have

$$\mathbb{E}_S\left[\mathbb{E}_s[|T|]\right] = \mathbb{E}_S[|T'|] \leq \mathbb{E}_{\tilde{S}}[|\tilde{T}'|] = n(n-1)\nu' \leq n^2\nu'.$$

By Markov inequality we have

$$\Pr\left(|T| \geq \frac{2\nu'}{\delta}n^2\right) \leq \frac{\delta}{2}. \tag{4}$$

Suppose $|T| < \frac{2\nu'}{\delta}n^2$ (which holds with probability $> 1 - \delta/2$). We now proceed to bound the number of labeling errors made by ADGAC. First, notice that in Algorithm 2, we divide all samples into groups of size $\alpha m = \varepsilon n/2$. For every set $S_i$, let

$$q(S_i) = \frac{1}{|S_i|} \min\left\{ \sum_{x \in S_i} I(h^*(x) = 1), \sum_{x \in S_i} I(h^*(x) = -1) \right\}$$

$$= \min\left\{ \Pr_{X \sim S_i}(h^*(x) = -1), \Pr_{X \sim S_i}(h^*(x) = 1) \right\}$$

where $X \sim S_i$ denote the empirical distribution that $X$ is drawn uniformly at random from the finite collection of points in $S_i$. Let

$$\beta = \frac{2}{\varepsilon}\sqrt{\frac{\nu'}{\delta}} \leq C\varepsilon^{\kappa-1}$$

for some constant $C$. Suppose $\varepsilon$ is small enough such that $\beta \leq 1/2$. Then we claim that there is at most 1 set $S_i$ such that $q(S_i) \geq \beta$. Otherwise, suppose two such sets exist; let them be $S_i$ and $S_j$. So there are at least $\alpha\beta m$ points $x \in S_i$ with $h^*(x) = -1$, and $\alpha\beta m$ points $x \in S_i$ with $h^*(x) = 1$; the same holds for $S_j$. These -1s and 1s would indicate at least

$$2\alpha^2\beta^2 m^2 = \frac{2\nu'}{\delta}n^2$$

anti-sort pairs, which violates our claim of $|T|$.

Since ADGAC uses group binary search, we first analyze some properties of the majority label of the Bayes optimal classifier within each group/set. For each set $S_i$, let $\mu(S_i) = \text{sign}(\sum_{x \in S_i} h^*(x_i))$ be the majority Bayes optimal label. We can show that $\mu(S_i)$ is monotonic: that is, for every $i < j$ we have $\mu(S_i) \leq \mu(S_j)$. To see this, suppose there exist two sets $S_i, S_j, i < j$ such that $\mu(S_i) = 1$ and $\mu(S_j) = -1$. That would indicate at least $\alpha^2 m^2/2 > \alpha^2\beta^2 m^2$ anti-sort pairs, which violates our assumption. So there must be a boundary $l$ such that $\mu(S_i) = -1$ for $i < l$, and $\mu(S_i) = 1$ for $i \geq l$. We call $S_l$ to be the *boundary set*. Now consider two cases:

- Case 1: there exists a set $S_{l'}$ such that $q(S_{l'}) \geq \beta$ (recall that from previous arguments, there is only one such set). If $l \neq l'$, the sets $S_l$ and $S_{l'}$ generates at least $2(\alpha m/2)(\alpha\beta m) \geq 2\alpha^2\beta^2 m^2$ anti-sort pairs, which violates our assumption for $|T|$. So $l = l'$.

- Case 2: for all sets $S_i$, we have $q(S_i) < \beta$.

In both cases, we have $q(S_i) < \beta$ for all $i \neq l$.

Now we prove that the majority vote of the noisy labels agrees with the majority vote of the Bayes optimal classifier $\mu(S_i)$ for each set $S_i$ that we visit, and hence we will find the boundary set $S_l$. Suppose $q(S_i) < \beta$. Take

$$t = \left(\frac{\varepsilon}{16\tilde{\mu}}\right)^{\kappa-1}.$$

For small enough $\varepsilon$, we have $t \leq 1/2$, and $\Pr(x : |\eta(x) - 1/2| \leq t) \leq \varepsilon/16$. Let $U = \{x_i \in S : |\eta(x_i) - 1/2| \leq t\}$. By relative form of Chernoff bound we have

$$\Pr\left(|U| > 3\log(4/\delta) + n\varepsilon/8\right) \leq \exp\left(-\frac{3\log(4/\delta) + \varepsilon n/16}{3}\right) \leq \frac{\delta}{4}.$$

Suppose $|U|/n \leq \varepsilon/8$, so at most 1/4 of each $S_i$ is in $U$.

For each set $S_i$, let $\bar{S}_i = \{x \in S_i : h^*(x) \neq \mu(S_i)\}$ and $S'_i = \{x \in S_i : |\eta(x) - 1/2| \leq t\}$. So for each set such that $q(S_i) \leq \beta$, we have

$$\Pr(Y \neq \mu(S_i)|X \sim S_i) \leq \Pr(Y \neq \mu(S_i)|X \in S'_i)\Pr(X \in S'_i|X \sim S_i) +$$
$$\Pr(Y \neq \mu(S_i)|X \in \bar{S}_i)\Pr(X \in \bar{S}_i|X \sim S_i) +$$
$$\Pr(Y \neq \mu(S_i)|X \notin S'_i, X \notin \bar{S}_i)\Pr(X \notin S'_i, X \notin \bar{S}_i|X \sim S_i)$$
$$\leq \left(\frac{1}{2} + t\right) \cdot \frac{1}{4} + 1 \cdot \beta + \left(\frac{1}{2} - t\right) \cdot \left(\frac{3}{4} - \beta\right)$$
$$= \frac{1}{2} - \frac{1}{2}t + \left(\frac{1}{2} + t\right)\beta.$$

Pick $\nu'$ small enough such that $\beta \leq \frac{1}{4}t$:

$$\frac{2}{\varepsilon}\sqrt{\frac{\nu'}{\delta}} \leq \frac{1}{4}\left(\frac{\varepsilon}{16\tilde{\mu}}\right)^{\kappa-1}.$$

This yields

$$\nu' \leq \frac{\varepsilon^{2\kappa}\delta}{32(16\tilde{\mu})^{2\kappa-2}}.$$

Note that this also guarantees $\beta \leq 1/2$ above, since $t \leq \frac{1}{2}$. Now we have

$$\Pr\left[Y \neq \mu(S_i)|X \sim S_i\right] \leq \frac{1}{2} - \frac{1}{2}t + \frac{1}{4}t\,(t + 1/2) \leq \frac{1}{2} - \frac{1}{4}t.$$

In the algorithm, suppose we pick $X_1, X_2, ..., X_k \in S_i$ as the points for which to query the label and the labels are $Y_1, ..., Y_k$. Note that since $n = \Omega\left(\left(\frac{1}{\varepsilon}\right)^{2\kappa-1}\log(1/\delta)\right)$, we have $|S_i| = \alpha n \geq k$ for every i, so we will not run out of samples for each set. By Hoeffding's inequality, we have

$$\Pr\left[\text{sign}\left(\sum_{j=1}^{k}Y_j\right) = \mu(S_i)\right] = \Pr\left[\frac{1}{k}\sum_{j=1}^{n}I(Y_j = \mu(S_i)) > \frac{1}{2}\right] \leq \exp\left(-\frac{1}{8}kt^2\right).$$

The choice of $k$ yields that the majority vote of the noisy labels agrees with the majority vote $\mu(S_i)$ of the Bayes optimal classifier for each $S_i$ with $q(S_i) \leq \beta$ we visit, with probability $\frac{\delta}{8\log(2/\varepsilon)}$.

Suppose the binary search output set $S_t$ (i.e., the value of $t$ at step 9). Now we analyze the errors we made in the final output. We consider the two cases:

- Case 1: If $q(S_l) \geq \beta$, then with probability $1 - \delta$, we have $t \in \{l-1, l, l+1\}$ since we might behave arbitrarily in set $S_l$. In this case, we have $q(S_l)|S_l| \geq \alpha\beta m$, and so $|\{x : x \in S_{t'}, t' < l, h^*(x) = 1\}| \leq \alpha\beta m$, because otherwise we have $\alpha^2\beta^2m^2$ anti-sort pairs, which violates our assumption on $|T|$. Similarly, $|\{x : x \in S_{t'}, t' > l, h^*(x) = -1\}| \leq \alpha\beta m$. Counting also the possible errors made on $S_l$, the total number of errors is

$$|\{\hat{y}_i : \hat{y}_i \neq h^*(x_i)\}| \leq \alpha m + 2\alpha\beta m \leq \frac{\varepsilon n}{2}\left(1 + \frac{1}{2}t\right) \leq \varepsilon n.$$

- Case 2: If $q(S_i) < \beta$ for all $i$, then we have $t \in \{l-1, l\}$. Now note that we have $|\{x \in S_{l-1} : h^*(x) = -1\}| \geq \alpha m/2$, and so $|\{x \in S_{t'} : t' < l-1, h^*(x) = 1\}| \leq \alpha\beta m$ since otherwise at least $\alpha^2\beta m/2$ anti-sort pairs are present. So $|\{x \in S_{t'} : t' \leq l-1, h^*(x) = 1\}| \leq 2\alpha\beta m$ considering $q(S_{l-1}) < \beta$. Similarly, $|\{x \in S_{t'} : t' \geq l, h^*(x) = -1\}| \leq 2\alpha\beta m$. So the total number of errors is
$$|\{\hat{y}_i : \hat{y}_i \neq h^*(x_i)\}| \leq 4\alpha\beta m \leq \varepsilon n.$$

So we have at most $\varepsilon n$ error under both cases. Now we examine the total query complexity: The expected comparison complexity is $O(m\log m)$ even if we have noisy comparisons, see [3]. For label complexity, it takes $k = \tilde{\mathcal{O}}\left(\log(1/\delta)\left(\frac{1}{\varepsilon}\right)^{2\kappa-2}\right)$ queries for each set $S_t$, and we do this for $\mathcal{O}(\log(1/\alpha)) = \mathcal{O}\left(\log\left(\frac{2m}{\varepsilon n}\right)\right)$ times. So the total query complexity is

$$\tilde{\mathcal{O}}\left(\log\left(\frac{2m}{\varepsilon n}\right)\log(1/\delta)\left(\frac{1}{\varepsilon}\right)^{2\kappa-2}\right).$$

$\square$

# E   Proof of Theorem 5

*Proof.* The first part of proof is exactly the same as that of Theorem 4. We now bound $\Pr[Y \neq \mu(S_i)|X \sim S_i]$. Suppose $q(S_i) < \beta$. Let $V = \{x : \Pr[Y \neq h^*(X)|X = x] > 1/4\}$ and $U = \{x_i : \Pr[Y \neq h^*(X)|X = x_i] > 1/4\}$. We have $P(V) \leq 4\nu$. By a relative Chernoff bound, if $\nu \leq C_1\varepsilon$ for a small enough constant $C_1$ we have

$$\Pr[|U| \leq 8n\nu + 3\log(4/\delta)] \leq \exp\left(-\frac{3\log(4/\delta) + 4\nu n}{3}\right) \leq \delta/4.$$

So if $\nu \leq \frac{1}{64}\varepsilon$ we have $|U|/n \leq \varepsilon/8$ with probability $\delta/4$. In this case, at most 1/4 of each $S_i$ is in $U$.

For each set $S_i$, let $\bar{S}_i = \{x \in S_i : h^*(x) \neq \mu(S_i)\}$ and $\tilde{S}_i = \{x \in S_i, x \in U\}$. So for each set such that $q(S_i) \leq \beta$, we have

$$
\begin{aligned}
\Pr(Y \neq \mu(S_i)|X \sim S_i) \leq &\Pr(Y \neq \mu(S_i)|X \in \tilde{S}_i)\Pr(X \in \tilde{S}_i|X \sim S_i)+ \\
&\Pr(Y \neq \mu(S_i)|X \in \bar{S}_i)\Pr(X \in \bar{S}_i|X \sim S_i)+ \\
&\Pr(Y \neq \mu(S_i)|X \notin \tilde{S}_i, X \notin \bar{S}_i)\Pr(X \notin \tilde{S}_i, X \notin \bar{S}_i|X \sim S_i) \\
\leq &1 \cdot \frac{1}{4} + 1 \cdot \beta + \left(\frac{3}{4} - \beta\right)\frac{1}{4} \\
= &\frac{7}{16} + \frac{3}{4}\beta.
\end{aligned}
$$

So there exists constant $C_2$ such that if $\nu' \leq C_2\varepsilon^2\delta$ we have $\beta \leq \frac{1}{24}$, $\Pr(Y \neq \mu(S_i)|X \sim S_i)] \leq \frac{1}{2} - \frac{1}{32}$. Thus by Hoeffding's inequality, the choice of $k$ yields that we recover $\mu(S_i)$ for each $i$ we visit with probability $\frac{\delta}{8\log(2/\varepsilon)}$.

By similar analysis as the proof of Theorem 4, we can show that the number of errors (i.e., $|\{\hat{y}_i : \hat{y}_i \neq h^*(x_i)\}|$) is at most $\varepsilon n$.

Now examine the total query complexity: It takes $k = \mathcal{O}\left(\log(\log(1/\varepsilon)/\delta)\right)$ queries for each set $S_t$, and we do this for $\mathcal{O}(\log(1/\alpha)) = \mathcal{O}(\log\left(\frac{2m}{\varepsilon n}\right))$ times. So the total query complexity is

$$\mathcal{O}\left(\log\left(\frac{2m}{\varepsilon n}\right)\log\left(\frac{\log(1/\varepsilon)}{\delta}\right)\right).$$

$\square$

# F   Proof for $A^2$-ADGAC

We use the following lemma adapted from [19]:

**Lemma 13** ([19], Lemma 3.1). *Suppose that $\mathcal{D} = \{x_1, x_2, ..., x_n\}$ is i.i.d. sampled from $\mathcal{P}_\mathcal{X}$, and $h^* \in \mathbb{C}$. There is a universal constant $c_0 \in (1, \infty)$ such that for any $\gamma \in (0, 1)$, and any $n \in \mathbb{N}$, letting*

$$U(n, \gamma) = c_0\frac{d\log(n/d) + \log(1/\gamma)}{n},$$

*with probability at least $1 - \gamma$, $\forall h \in \mathbb{C}$, the following inequalities hold:*

$$\Pr_{X \sim \mathcal{P}_\mathcal{X}}[h(X) \neq h^*(X)] \leq \max\{2\Pr_{X \sim \mathcal{D}}[h(X) \neq h^*(X)], U(n, \gamma)\},$$

$$\Pr_{X \sim \mathcal{D}}[h(X) \neq h^*(X)] \leq \max\{2\Pr_{X \sim \mathcal{P}_\mathcal{X}}[h(X) \neq h^*(X)], U(n, \gamma)\}.$$

*Here $X \sim \mathcal{D}$ means $X$ is uniformly sampled from finite set $\mathcal{D}$.*

*Proof of Theorem 6.* For a labeled dataset $W = \{(x_i, \hat{y}_i)\}_{i=1}^n$, let $\mathsf{err}_W(h) = \frac{1}{n}\sum_{i=1}^n I(h(x_i) \neq \hat{y}_i)$ be the empirical risk of $h$ on $W$ for any $h \in \mathbb{C}$ (remind that $\hat{y}_i$ are predictions of ADGAC). For a clearer explanation, we formalize the $A^2$-ADGAC algorithm in Algorithm 3. We use induction to

---
**Algorithm 3** $A^2$-ADGAC
---
**Input:** $n_i, \mathbb{C}, \varepsilon, \delta$, comparison oracle $f$.
 1: Let $V \leftarrow \mathbb{C}$.
 2: **for** $i = 1, 2, ..., \lceil \log(1/\varepsilon) \rceil$ **do**
 3:     Sample dataset $\tilde{S}$ of size $n_i$.
 4:     Let $S \leftarrow \{x \in \tilde{S} : x \in \mathsf{DIS}(V)\}$.
 5:     Run ADGAC (Subroutine 2) with $S, \tilde{S}, \varepsilon_i = 2^{-(i+2)}, k_i$ and labeled dataset $W$.
 6:     $V = V \setminus \{h : |W| \mathsf{err}_W(h) \geq n_i \varepsilon_i\}$.
**Output:** Any Classifier $\hat{h} \in V$.
---

prove that after iteration $i$ we have $\Pr[h(X) \neq h^*(X)] \leq 4\varepsilon_i$ for all $h \in V$ after step 6 in Algorithm 3. This proposition holds for $i = 0$. Suppose it holds for $i - 1$. By Theorem 4 and a union bound, with probability $1 - \delta/4$, for every iteration $i$ we have at most $n_i \varepsilon_i$ errors with respect to $h^*$ after running ADGAC, i.e., $|W| \mathsf{err}_W(h^*) \leq n_i \varepsilon_i$. So $h^*$ will not be eliminated from $V$ in any iteration with probability $1 - \delta/4$. On the other hand, notice that by Step 6 in Algorithm 3 all functions $h \in V$ satisfies $|W| \mathsf{err}_W(h) \leq n_i \varepsilon_i$, so by triangle inequality we have (notice that $W$ is just the set $S$ with labels)

$$|S| \Pr_{X \sim S}[h(X) \neq h^*(X)] = |\{x \in S : h(x) \neq h^*(x)\}|$$
$$\leq |\{(x, \hat{y}) \in W : h(x) \neq \hat{y}\}| + |\{(x, \hat{y}) \in W : h^*(x) \neq \hat{y}\}|$$
$$\leq 2\varepsilon_i n_i.$$

Also note that functions in $V$ agrees on $\tilde{S} \setminus S$; so $|\tilde{S}| \Pr_{X \sim \tilde{S}}[h(X) \neq h^*(X)] \leq 2\varepsilon_i n_i$, and since $|\tilde{S}| = n_i$ we have $\Pr_{X \sim \tilde{S}}[h(X) \neq h^*(X)] \leq 2\varepsilon_i$. Now using Lemma 13 with $n = n_i$, we have $\Pr_{X \sim \mathcal{P}_X}[h(x) \neq h^*(x)] \leq 4\varepsilon_i$ for every $h \in V$ by choosing $n_i$ such that $U\left(n_i, \frac{\delta}{4\log(1/\varepsilon)}\right) \leq \varepsilon_i$. So at the end of the algorithm it outputs a classifier with $\Pr[\hat{h} \neq h^*] \leq \varepsilon$.

Now we examine the number of queries. By definition of disagreement coefficient, at round $i$ we have $\mathsf{DIS}(V) \leq \theta \varepsilon_i$; thus using a relative Chernoff bound we know that with probability $1 - \delta/4$ we have

$$m_i := |S| \leq \log(12/\delta) + 2n_i \theta \varepsilon_i = \mathcal{O}\left(\theta\left((d\log(1/\varepsilon)) + \left(\frac{1}{\varepsilon_i}\right)^{2\kappa-2} \log(1/\delta)\right)\right).$$

It takes $O(m_i \log m_i)$ comparisons in expectation to rank the set, and there are $\log(1/\varepsilon)$ iterations. So the total comparison complexity is

$$\mathbb{E}[\mathsf{SC}_{\mathrm{comp}}] = \tilde{\mathcal{O}}\left(\theta\log\left(\frac{1}{\varepsilon}\right)\left(\log d\theta + (\kappa-1)\log\left(\frac{1}{\varepsilon}\right)\right)\left(\left(d\log\left(\frac{1}{\varepsilon}\right)\right) + \left(\frac{1}{\varepsilon}\right)^{2\kappa-2}\log(1/\delta)\right)\right).$$

This obtained the stated comparison complexity. The label complexity follows by multiplying the label complexity of ADGAC by $\log(1/\varepsilon)$. Note that in every step we have $\frac{m_i}{\varepsilon_i n_i} = O\left(\min\left\{\theta, \frac{1}{\varepsilon}\right\}\right)$. $\qquad\square$

*Proof of Theorem 7.* With the same proof, $A^2$-ADGAC outputs a classifier with $\Pr[\hat{h} \neq h^*] \leq \varepsilon$ upon finishing. We know examine the number of queries. By definition of disagreement coefficient, at round $i$ we have $\mathsf{DIS}(V) \leq \theta \varepsilon_i$; thus using a Chernoff bound we know that with probability $1 - \delta/4$ we have

$$m_i := |S| \leq \log(12/\delta) + 2n_i \theta \varepsilon_i = \mathcal{O}\left(\theta d\log\left(\frac{1}{\varepsilon_i}\right)\log\left(\frac{1}{\delta}\right)\right).$$

It takes $O(m_i \log m_i)$ comparisons in expectation to rank the set, and there are $\log(1/\varepsilon)$ iterations. So the total comparison complexity is

$$\mathbb{E}[\mathsf{SC}_{\mathrm{comp}}] = \tilde{\mathcal{O}}\left(\theta d\log(\theta d)\log\left(\frac{1}{\varepsilon_i}\right)\log\left(\frac{1}{\delta}\right)\right).$$

This obtained the stated comparison complexity. The label complexity follows by multiplying the label complexity of ADGAC by $\log(1/\varepsilon)$. Note that in every step we have $\frac{m_i}{\varepsilon_i n_i} = O\left(\min\left\{\theta, \frac{1}{\varepsilon}\right\}\right)$. $\qquad\square$

# G  Proof for Margin-ADGAC

---

**Algorithm 4** Margin-ADGAC: Efficiently learning halfspaces with comparison

---

**Input:** $\varepsilon, \delta$, target errors $\varepsilon_k$, sample sizes $n_k$, sequences $r_k, b_k, \tau_k$, precision value $\kappa$.

1: Draw $n_1$ unlabeled samples to $S$ and run ADGAC with $\left(S, n_1, \varepsilon_0, \frac{\delta}{8\log(1/\varepsilon)}, k_1\left(\varepsilon_0, \frac{\delta}{8\log(1/\varepsilon)}\right)\right)$, and obtain a labeled dataset $W$.

2: **for** $k = 1, 2, ..., s = \lceil \log(4/\varepsilon) \rceil$ **do**

3:    Find $v_k \in B(w_{k-1}, r_k)$ that approximately minimize training hinge loss over $W$, with $\|v_k\|_2 \leq 1$:
$$l_{\tau_k}(v_k, W) \leq \min_{w \in B(w_{k-1}, r_k) \cap B(0,1)} l_{\tau_k}(w, W) + \kappa/8.$$

4:    $w_k \leftarrow \frac{v_k}{\|v_k\|_2}$.

5:    Sample another dataset $\tilde{S}$ of $n_k$ unlabeled samples.

6:    $S = \{x \in S : |w_k \cdot x| \leq b_k\}$.

7:    Run ADGAC with $\left(S, n_k, \varepsilon_k, \frac{\delta}{8\log(1/\varepsilon)}, k^{(1)}\left(\varepsilon_k, \frac{\delta}{8\log(1/\varepsilon)}\right)\right)$ and obtain labeled dataset $W$.

**Output:** Return $w_s$.

---

We first prove Theorem 8, and Theorem 9 follows exactly the same proof with $\kappa = 1$ and using Theorem 5. For clearer explanation, we re-illustrate Margin-ADGAC in a form similar to that in [6] in Algorithm 4. The proof mostly follows that of [6]. We give a refined sample complexity via Rademacher complexity following the ideas in [32], and also change the proof according to the properties of ADGAC (note that we are not using independent samples by replace the sampling step with ADGAC).

To simplify notations, let $\mathsf{err}(w)$ be $\mathsf{err}(h_w(x)) = \mathsf{err}(\mathrm{sign}(w \cdot x))$. Define $\Delta_D(w, w') = \Pr_{X \sim D}[\mathrm{sign}(w \cdot X) \neq \mathrm{sign}(w' \cdot X)]$. Also, let $\theta(w_1, w_2)$ be the angle between two vectors $w_1, w_2$. Let $D_{w,\gamma} = \{x : |w \cdot x| \leq \gamma\}$.

The key step is to prove the following theorem:

**Theorem 14.** *For $k \leq \log(1/\varepsilon)$, if $\Delta_{\mathcal{P}_\mathcal{X}}(w_{k-1}, w^*) \leq M^{-(k-1)}$, with probability $1 - \frac{\delta}{k+k^2}$, after round $k$ of Margin-ADGAC we have $\Delta_{D_{w_{k-1}, b_{k-1}}}(w_k, w^*) \leq \kappa$.*

To prove the theorem, we first list useful properties of isotropic log-concave distributions and fix the parameters we use for the algorithm. We use exactly the same parameters for $r_i, \tau_i, b_i, z_i$ as in [6], and we restate them here for completeness.

**Lemma 15** ([6, 10, 25]). *Suppose $X \sim \mathcal{P}_\mathcal{X}$ is a isotropic log-concave distribution in $\mathbb{R}^d$ with probability density function $f$. Then*

    *(a) There is an absolute constant $c_1$ such that, if $d = 1$, $f(x) > c_1$ for all $x \in [-1/9, 1/9]$.*

    *(b) There is an absolute constant $c_2$ such that for any two unit vectors $u$ and $v$ in $R^d$ we have $c_2\theta(u, v) \leq \Delta_{\mathcal{P}_\mathcal{X}}(u, v)$.*

    *(c) There exists constant $c_3$ such that for any unit vector $w$ and $\gamma > 0$, $\Pr[|w \cdot X| \leq \gamma] \leq c_3\gamma$.*

    *(d) There is a constant $c_4$ such that for any unit vector $u$, all $0 < \gamma < 1$, for all $a$ such that $\|u - a\|_2 \leq \gamma$ and $\|a\|_2 \leq 1$, $\mathbb{E}_{X \sim D_{u,\gamma}}[(a \cdot X)^2] \leq c_4(r^2 + \gamma^2)$.*

    *(e) For any $c_5 > 0$, there is a constant $c_6 > 0$ such that the following holds: let $u$ and $v$ be two unit vectors in $\mathbb{R}^d$, and assume that $\theta(u, v) = \alpha < \pi/2$. Then $\Pr_{X \sim \mathcal{P}_\mathcal{X}}[\mathrm{sign}(u \cdot X) \neq \mathrm{sign}(v \cdot X)$ and $|v \cdot X| \leq c_6\alpha] \leq c_5\alpha$.*

Now we give the settings of parameters. Let $M = \max\{\frac{2}{c_2\pi}, 2\}$. Let $c_1'$ be the value of $c_6$ in Lemma 15 corresponding to the case where $c_5$ is $\frac{c_2}{4M}$; let $b_k = c_1'M^{-k}$. Let $r_k = \min\{M^{-(k-1)}/c_2, \pi/2\}$ and $\kappa = \frac{1}{4c_1'M}$. Let $\tau_k = \frac{c_1 \min\{b_{k-1}, 1/9\}\kappa}{6}$, and $z_k^2 = r_k^2 + b_{k-1}^2$. Let $\varepsilon_k = \frac{c_3\tau_k^2 b_k \kappa^2}{256c_4 z_k^2}$, and $n_k = O\left(\frac{1}{b_k}d\log^3\left(\frac{dk}{1/\delta}\right)\right)$. Also let $m_k = 2c_3 b_k n_k + \log(12k/\delta)$.

Then we prove the following lemma:

**Lemma 16.** *Suppose $|W| \geq m_k$. Let $c(W)$ be the set with truthful labels w.r.t. $w^*$, i.e., $c(W) = \{(x, \text{sign}(w^* \cdot x)) : x \in W\}$. For any $w \in B(w_{k-1}, r_k)$, with probability $1 - \frac{\delta}{3(k+k^2)}$ we have*

$$|l(w, W) - l(w, c(W))| \leq \kappa/8.$$

*Proof.* Let $N = \{(x, \hat{y}) \in W : \hat{y} \neq \text{sign}(w^* \cdot x)\}$ be the set where ADGAC has $x$'s label different than $\text{sign}(w^* \cdot x)$ (remind that $\hat{y}$ is the prediction of ADGAC). We have

$$l(w, W) = \frac{1}{|W|} \sum_{(x,\hat{y}) \in W} l_{\tau_k}(w, x, \hat{y})$$

$$= \frac{1}{|W|} \left( \sum_{(x,y) \notin N} l_{\tau_k}(w, x, \text{sign}(w^* \cdot x)) + \sum_{(x,y) \in N} l_{\tau_k}(w, x, -\text{sign}(w^* \cdot x)) \right).$$

So

$$|l(w, W) - l(w, c(W))| \leq \frac{1}{\tau_k |W|} \sum_{x \in N} 2(w \cdot x)$$

$$\leq \frac{1}{\tau_k |W|} \sum_{x \in W} I(x \in N) 2(w \cdot x). \tag{5}$$

We use the following lemma from [6]:

**Lemma 17** (Lemma D.4, [6]). *For an absolute constant $c$, with probability $1 - \frac{\delta}{6(k+k^2)}$,*

$$\max_{x \in W} \|x\|_2 \leq c\sqrt{d} \log\left(\frac{|W|k}{\delta}\right).$$

Note that
$$|w \cdot x| \leq |w_{k-1} \cdot x| + |(w - w_{k-1}) \cdot x| \leq b_k + r_k \|x\|_2.$$
So with probability $1 - \frac{\delta}{6(k+k^2)}$, an event $E_\delta$ happens such that

$$\frac{|w \cdot x|}{\tau_k} \leq c'\sqrt{d} \log\left(\frac{|W|k}{\delta}\right)$$

for all $x \in W$, for some constant $c'$.

Notice that $\frac{|N|}{|W|} \leq \frac{\varepsilon_k n_k}{m_k}$. Let $N'$ be a $\frac{\varepsilon_k n_k}{m_k}$ fraction of $W$ with the largest values of $|w \cdot x|$. Let $\varphi(W) = \sum_{x \in N'} |w \cdot x|$. So by (5) we have $|l(w, W) - l(w, c(W))| \leq \frac{2}{\tau_k |W|} \varphi(W)$. Now we have

$$\mathbb{E}[\varphi(W)] = \mathbb{E}\left[\sum_{x \in W} \delta(x \in N') |w \cdot x|\right]$$

$$\leq \sqrt{\frac{|N'|}{|W|}} \mathbb{E}\left[\sqrt{\sum_{x \in W} (w \cdot x)^2}\right]$$

$$\leq \sqrt{\frac{\varepsilon_k n_k}{m_k}} \sqrt{\mathbb{E}\left[\sum_{x \in W} (w \cdot x)^2\right]}$$

$$\leq \sqrt{\frac{\varepsilon_k n_k}{m_k}} \sqrt{c_4 z_k |W|} \leq \kappa \tau_k |W|/16.$$

The first inequality is by Cauchy-Schwartz inequality; the second is by Jensen's inequality; the third inequality is by property (d) in Lemma 15; the last inequality is by the value of $\varepsilon_i$. If we condition $\mathcal{P}_\mathcal{X}$ on $E_\delta$, the above expectation will be smaller since we bound $|w \cdot x|$ from above. Now by Mcdiarmid's

inequality, $\frac{1}{|W|}\varphi(W)$ deviates by at most $\frac{c'\sqrt{d}\log\left(\frac{|W|k}{\delta}\right)}{|W|}$ when we change a single value of $w \cdot x$ for some $x \in W$. So by McDiarmid's inequality, using $|W| \geq m_k = \Omega(d\log^2(d/\delta))$, with probability $1 - \frac{\delta}{3(k+k^2)}$ we have

$$|l(w,W) - l(w,c(W))| \leq \mathbb{E}[\varphi(W)|E_\delta] + \kappa/16 \leq \kappa/8.$$

$\square$

The other lemma is about bounding the difference between $l(w,c(W))$ and $E_W[l(w,c(W))]$. We improve the results in [6] using Rademacher complexity as below.

**Lemma 18.** *With probability $1 - \frac{\delta}{6(k+k^2)}$ we have*

$$|\mathbb{E}_W[l(w,x,sign(w^* \cdot x))] - l(w,W)| \leq \kappa/16.$$

*Proof.* Note that every $x \in W$ is sampled independently from $D_{w_k,b_{k-1}}$. Following the same proof as in Lemma 16, an Event $E_\delta$ happens with probability $1 - \frac{\delta}{6(k+k^2)}$ that

$$\left|\frac{w \cdot x}{\tau_k}\right| \leq c'\sqrt{d}\log\left(\frac{|W|k}{\delta}\right)$$

for all $x \in W$, for some constant $c'$. This means $l_{\tau_k}(w,x,\text{sign}(w^* \cdot x))$ are also bounded in the same range under $E_\delta$.

Define the function class $\mathcal{F} = \{x \to l_{\tau_k}(w,x,\text{sign}(w^* \cdot x)), \|w - w_k\| \leq r_k\}$. On event $E_\delta$, all functions in $\mathcal{F}$ are bounded. Now we bound the Rademacher complexity $R_n(\mathcal{F})$. Actually, define $\mathcal{F}' = \{x \to \frac{1}{\tau_k}w \cdot x \cdot \text{sign}(w^* \cdot x), \|w - w_k\| \leq r_k\}$, we have $R_n(\mathcal{F}) \leq R_n(\mathcal{F}')$ by contraction inequality of Rademacher complexity (since hinge loss is 1-Lipschitz). So

$$R_n(\mathcal{F}) \leq R_n(\mathcal{F}')$$

$$= \frac{1}{\tau_k n}E_{x_1,\ldots,x_n \sim D_{w_k,b_{k-1}}}E_{\sigma_1,\ldots,\sigma_n}\sup_{w:\|w-w_k\|\leq r_k}\sum_{i=1}^{n}\sigma_i w \cdot x_i \cdot \text{sign}(w^* \cdot x_i)$$

$$= \frac{1}{\tau_k n}E_{x_1,\ldots,x_n \sim D_{w_k,b_{k-1}}}E_{\sigma_1,\ldots,\sigma_n}\sup_{w:\|w-w_k\|\leq r_k}\sum_{i=1}^{n}\sigma_i (w \cdot x_i) \qquad (6)$$

$$= \frac{1}{\tau_k n}E_{x_1,\ldots,x_n \sim D_{w_k,b_{k-1}}}E_{\sigma_1,\ldots,\sigma_n}\sum_{i=1}^{n}\sigma_i (w_k \cdot x_i)+$$

$$\frac{1}{\tau_k n}E_{x_1,\ldots,x_n \sim D_{w_k,b_{k-1}}}E_{\sigma_1,\ldots,\sigma_n}\sup_{w:\|w-w_k\|\leq r_k}\sum_{i=1}^{n}\sigma_i (w-w_k) \cdot x_i$$

$$= \frac{1}{\tau_k n}E_{x_1,\ldots,x_n \sim D_{w_k,b_{k-1}}}E_{\sigma_1,\ldots,\sigma_n}\sup_{w:\|w-w_k\|\leq r_k}\sum_{i=1}^{n}\sigma_i (w-w_k) \cdot x_i$$

$$= \frac{1}{\tau_k n}E_{x_1,\ldots,x_n \sim D_{w_k,b_{k-1}}}E_{\sigma_1,\ldots,\sigma_n}\sup_{w:\|w-w_k\|\leq r_k}(w-w_k)\sum_{i=1}^{n}\sigma_i \cdot x_i$$

$$\leq \frac{1}{\tau_k n}E_{x_1,\ldots,x_n \sim D_{w_k,b_{k-1}}}E_{\sigma_1,\ldots,\sigma_n}\sup_{w:\|w-w_k\|\leq r_k}\|w-w_k\|_2\left\|\sum_{i=1}^{n}\sigma_i x_i\right\|_2$$

$$\leq \frac{2r_k}{\tau_k n}\sqrt{E_{x_1,\ldots,x_n \sim D_{w_k,b_{k-1}}}E_{\sigma_1,\ldots,\sigma_n}\left\|\sum_{i=1}^{n}\sigma_i x_i\right\|_2^2} \qquad (7)$$

$$\leq \frac{2r_k}{\tau_k n}\sqrt{E_{x_1,\ldots,x_n \sim D_{w_k,b_{k-1}}}E_{\sigma_1,\ldots,\sigma_n}\left[\sum_{i=1}^{n}\|x_i\|_2^2 + \sum_{i,j}\sigma_i\sigma_j x_i \cdot x_j\right]} \qquad (8)$$

$$\leq \mathcal{O}\left(\frac{1}{n} \cdot \sqrt{nd\log^2\left(\frac{nk}{\delta}\right)}\right) \tag{9}$$

$$= \mathcal{O}\left(\sqrt{\frac{d\log^2\left(\frac{nk}{\delta}\right)}{n}}\right).$$

(6) is by the property that $\sigma_i \cdot \text{sign}(w^* \cdot x_i)$ has the same distribution as $\sigma_i$, and thus we can substitute $\sigma_i \cdot \text{sign}(w^* \cdot x_i)$ with a single variable; (7) is by Jensen's inequality, and (9) is by the boundary condition on $\|x\|_2$. So by Rademacher's inequality we have

$$|\mathbb{E}_W[l(w, x, \text{sign}(w^* \cdot x))] - l(w, W)| \leq R_{|W|}(\mathcal{F}) + \sqrt{\frac{\log(1/\delta)}{|W|}} C\sqrt{d}\log\left(\frac{|W|k}{\delta}\right)$$

$$\leq \mathcal{O}\left(\sqrt{\frac{d\log^2\left(\frac{|W|k}{\delta}\right)}{|W|}}\right) + \sqrt{\frac{\log(k/\delta)}{|W|}} C\sqrt{d}\log\left(\frac{|W|k}{\delta}\right)$$

$$= \mathcal{O}\left(\sqrt{\frac{d\log(k/\delta)}{|W|}}\log\left(\frac{|W|k}{\delta}\right)\right).$$

The choice of $|W| = m_i = \Omega\left(d\log^3\left(\frac{dk}{1/\delta}\right)\right)$ makes the above quantity less than $\kappa/16$. $\qquad\square$

Now we are ready to prove Theorem 14.

*Proof of Theorem 14.* With a probability of $1 - \frac{\delta}{k+k^2}$, suppose the conditions in Lemma 16 and 18 holds for $w = v_k$ and $w = w^*$. We have

$$\Delta_{D_{w_{k-1}, b_{k-1}}}(w_k, w^*)$$
$$= \Delta_{D_{w_{k-1}, b_{k-1}}}(v_k, w^*)$$
$$\leq \mathbb{E}_{x \in D_{w_{k-1}, b_{k-1}}}[l(v_k, x, \text{sign}(w^* \cdot x))] \qquad \text{(Since hinge loss upper bounds 0-1 loss)}$$
$$\leq l(v_k, c(W)) + \kappa/16 \qquad \text{(Using Lemma 18)}$$
$$\leq l(v_k, W) + \kappa/8 \qquad \text{(Using Lemma 16)}$$
$$\leq l(w^*, W) + \kappa/4 \qquad \text{(By the process of selecting } v_k)$$
$$\leq l(w^*, c(W)) + \kappa/4 + \kappa/16 \qquad \text{(Using Lemma 16)}$$
$$\leq L(w^*) + \kappa/4 + \kappa/8 \qquad \text{(Using Lemma 18)}$$
$$\leq \kappa. \qquad \text{(Using Lemma 3.7 in [6])}$$
$$\qquad\square$$

Now we can prove Theorem 8.

*Proof of Theorem 8.* By relative Chernoff bound and property (c) in Lemma 15, with probability $1 - \frac{\delta}{6(k+k^2)}$ we have $|W| \geq m_k = 2c_3 b_k n_k + \log(12k/\delta)$ in every iteration. Then the correctness of Margin-ADGAC follows the same way as in [6]. Now examine the number of queries: In each step we need to compare $m_i$ instances, as well as fitting the minimum requirement of ADGAC. So the comparison complexity is

$$\mathbb{E}[\text{SC}_{\text{comp}}] = \tilde{\mathcal{O}}\left(\log^2(1/\varepsilon)\left(d\log^4(d/\delta) + \left(\frac{1}{\varepsilon}\right)^{2\kappa-2}\log(1/\delta)\right)\right).$$

The label complexity is again obtained by multiplying the label complexity in each iteration by $\log(1/\varepsilon)$. Note that $\frac{\varepsilon_k n_k}{m_k}$ is constant in each iteration. Therefore,

$$\text{SC}_{\text{label}} = \tilde{\mathcal{O}}\left(\log(1/\varepsilon)\log(1/\delta)\left(\frac{1}{\varepsilon}\right)^{2\kappa-2}\right).$$

$$\qquad\square$$

*Proof of Theorem 9.* The proof follows exactly the same process as that of Theorem 8 using $\kappa = 1$, and Theorem 5. □

# H   Proof of Lower Bounds

## H.1   Proof of Theorem 10

*Proof.* Suppose $g(x_1) = a$ and $g(x_0) = b$ for $x_1, x_2 \in \mathcal{X}, a < b$. Let $h_1(x) = \text{sign}(g(x) - a)$ and $h_2(x) = \text{sign}(g(x) - b)$. Note that using $Z(x_1, x_2) = 0$ incurs $\nu' = 0$ for both $h^* = h_1$ and $h^* = h_2$, and thus comparison cannot distinguish between $h_1$ and $h_2$. Suppose $\mathbb{C} = \{h_1, h_2\}$. Thus, any algorithm $\mathcal{A}$ using both comparison and labeling oracles can be transformed into an algorithm $\mathcal{A}'$ that uses labeling oracle only, by making the comparison oracle always return 0. Note that $\mathsf{SC}_{\text{label}}(\mathcal{A}) = \mathsf{SC}_{\text{label}}(\mathcal{A}')$, so we only need to lower bound $\mathsf{SC}_{\text{label}}(\mathcal{A}')$. In the following, we adapt the proof in [19] to give a lower bound. The main difference is that our goal is to reach a small $\Pr[h(X) \neq h^*(X)]$, whereas in [19] the goal is a small $\mathsf{err}(h) - \mathsf{err}(h^*)$.

Let $P(x_1) = 24\varepsilon, P(x_0) = 1 - 24\varepsilon$. Consider two distributions $\mathcal{P}_1, \mathcal{P}_2$ over $\mathcal{X} \times \mathcal{Y}$ with two different Bayes function $\eta_1(), \eta_2()$. Let $\gamma = \varepsilon^{\kappa-1}$ if $\kappa > 1$, or $\gamma = \frac{1}{48}$ if $\kappa = 1$. Let $\eta_1(x_0) = \eta_2(x_0) = 1$, $\eta_1(x_1) = \frac{1}{2} + \gamma, \eta_2(x_1) = \frac{1}{2} - \gamma$. It is easy to verify both $\mathcal{P}_1$ and $\mathcal{P}_2$ satisfy Tsybakov noise condition.

Choose the groundtruth distribution to be $\mathcal{P}_1$ or $\mathcal{P}_2$ both with probability $1/2$. By the same proof as Theorem 4.3 in [19], an event happens with probability at least $\delta$ that $\hat{h}(x_1) \neq h^*(x_1)$, and thus $\Pr[\hat{h}(X) \neq h^*(X)] \geq \varepsilon$, if at most $2\lfloor \frac{1-\gamma^2}{\gamma^2} \log\left(\frac{1}{8\delta(1-2\delta)}\right)\rfloor$ labels are queried. So we prove the theorem for TNC.

The proof for adversarial noise is the same as the above proof using $\kappa = 1$. □

## H.2   Proof of Theorem 11

*Proof of Theorem 11.* The first term in (2) follows directly from Theorem 10. For the second term, we consider the case where both labeling and comparison oracles are perfect with $\nu = \nu' = 0$. This is a special case for all Conditions 1, 2 and 3. Notice that in this case, a perfect comparison oracle can be constructed from a labeling oracle by $Z(x, x') = \text{sign}(Y(x) - Y(x')) = \text{sign}(h^*(x) - h^*(x'))$; thus, any algorithm $\mathcal{A}$ with access to both labeling and comparison oracles can be transformed into another algorithm $\mathcal{A}'$ that uses labeling oracle (by replacing the comparison oracle with one that queries labeling oracle instead). So we have

$$2\mathsf{SC}_{\text{comp}}(\mathcal{A}) + \mathsf{SC}_{\text{label}}(\mathcal{A}) = \mathsf{SC}_{\text{label}}(\mathcal{A}') = \Omega(d\log(1/\varepsilon)),$$

where $\Omega(d\log(1/\varepsilon))$ is the standard lower bound for realizable active learning (see e.g., [19]). □

## H.3   Proof of Theorem 12

Define $R^B(\hat{g})$ to be the error of comparison oracle induced by $\hat{g}$, and also $\mathbb{C}_{\hat{g}} = \{h : h(x) = \text{sign}(\hat{g}(x) - t), t \in \mathbb{R}\}$. To prove Theorem 12, we first give a lower bound on the left hand side (Theorem 19) by giving a $\hat{g}$ that every $h \in \mathbb{C}_{\hat{g}}$ will have every at least $\sqrt{\nu'}$. Then we give an upper bound on it (Theorem 20) by finding a good estimator $t$. We find $t$ by reducing $\Pr[\text{sign}(\hat{g}(X) - t) \neq h^*(X)]$ to the case when for every $x, x'$ such that $\hat{g}(x) = \hat{g}(x')$ we also have $h^*(x) = h^*(x')$. We find such a good function $f$ in this case by fixing the amount of error at each value of $\hat{g}(x)$, and carefully adjusting the noise levels.

**Theorem 19.** *Suppose* $\min\{\Pr[h^*(X) = 1], \Pr[h^*(X) = -1]\} \geq \sqrt{\nu'}$. *For any $g^*$ such that $g^*(X)$ has a density function, there exists $\hat{g}$ which induces a comparison oracle with error $\nu'$, such that for every $h \in \mathbb{C}_{\hat{g}}$, we have* $\Pr[h(X) \neq h^*(X)] \geq \sqrt{\nu'}$.

*Proof.* Consider the distribution of $g^*(X)$. Pick a consecutive interval $I = [a, b]$ with $a < 0 < b$ such that $\Pr(g^*(X) \in [0, b]) = \Pr(g^*(X) \in [a, 0]) = \sqrt{\nu'}$. Pick some integer $n \in \mathbb{N}$. Suppose the

cdf and pdf of random variable $T = g^*(X)$ is $F(t)$ and $p(t)$ respectively. Define

$$\hat{g}(x) = \begin{cases} a + (b-a)\frac{F(g^*(x))-F(a)}{\sqrt{v'}}, & \text{if } x \in [a, 0], \\ a + (b-a)\frac{F(g^*(x))-F(0)}{\sqrt{v'}}, & \text{if } x \in (0, b], \\ g^*(x), & \text{otherwise.} \end{cases}$$

The error of the comparison oracle induced by $\hat{g}$ can be represented as

$$R^B(\hat{g}) = 2 \int_{g^*(x)\in(0,b]} p(g^*(x)) \int_{g^*(x')\in[a,0)} p(g^*(x')) \cdot \delta(\hat{g}(x') > \hat{g}(x)) \, \mathrm{d}g^*(x)\mathrm{d}g^*(x')$$

Let $t = g^*(x)$ and $t' = g^*(x')$. Then $\hat{g}(x') > \hat{g}(x)$ if and only if

$$F(t') - F(a) > F(t) - F(0),$$
$$\Leftrightarrow F(t) - F(t') < \sqrt{\nu'}.$$

For every $t \in [0, b]$, let $G(t)$ satisfy $F(t) - F(G(t)) = \sqrt{\nu'}$. Then

$$
\begin{aligned}
R^B(\hat{g}) &= 2 \int_{t=0}^{b} p(t) \int_{t'=a}^{0} p(t') \cdot \delta\left(F(t) - F(t') < \sqrt{\nu'}\right) \, \mathrm{d}t\mathrm{d}t' \\
&= 2 \int_{t=0}^{b} p(t) \int_{t'=a}^{G(t)} p(t') \, \mathrm{d}t\mathrm{d}t' \\
&= 2 \int_{t=0}^{b} p(t)(F(G(t)) - F(a))\mathrm{d}t \\
&= 2 \int_{t=0}^{b} p(t)(F(t) - F(0))\mathrm{d}t \\
&= 2 \int_{t=0}^{b} p(t) \int_{t'=0}^{t} p(t')\mathrm{d}t\mathrm{d}t' \\
&= 2 \int_{t=0}^{b} \int_{t'=0}^{b} p(t)p(t')\delta(t' < t)\mathrm{d}t\mathrm{d}t' \\
&= \nu'.
\end{aligned}
$$

Now examine any function in $\mathbb{C}_{\hat{g}}$. If we pick a threshold $t \notin [a, b]$, the error is at least $\sqrt{\nu'}$ since we incur error on either $\{x : g^*(x) \in [a, 0]\}$ or $\{x : g^*(x) \in [0, b]\}$. If we pick threshold $a + (b-a)t$ for $t \in [0, 1]$, we induce an error for any $g^*(x) \in [a, 0]$ with $\frac{F(g^*(x))-F(a)}{\sqrt{v}} > t$, and any $g^*(x) \in (0, b]$ with $\frac{F(g^*(x))-F(0)}{\sqrt{v}} < t$. A routine calculation shows the error is always $\sqrt{\nu'}$.

$\square$

**Theorem 20.** *Suppose that $\hat{g}$ induces a comparison oracle with error $\nu'$, and also distributions of $\hat{g}(X)$ and $g^*(X)$ are smooth in the sense that they both have a density function. There exists $h_t(x) := sign(\hat{g}(x) - t) \in \mathbb{C}_{\hat{g}}$ such that the error of $h_t(x)$ with respect to $h^*(x)$ is at most $\sqrt{\nu'}$, i.e.,*

$$\Pr[h_t(X) \neq h^*(X)] = \Pr[(\hat{g}(X) - t)g^*(X) < 0] \leq \sqrt{\nu'}.$$

We first prove the inequality:

**Lemma 21.** *Suppose $\{x_i\}_{i=1}^n$ and $\{y_i\}_{i=1}^n$ satisfies $x_i, y_i \in \mathbb{R}, x_i, y_i \geq 0$. If $\sum_{i=1}^n \sum_{j=i}^n x_i y_j \leq t$, we have*

$$\min_{k=0,1,\dots,n} \{x_1 + \cdots + x_k + y_{k+1} + \cdots + y_n\} \leq \sqrt{\frac{2nt}{n+1}},$$

*the equality holds when $x_1 = x_2 = \cdots = x_n = y_1 = \cdots = y_n = \sqrt{\frac{2t}{n(n+1)}}$.*

*Proof of Lemma 21.* Let $f(k) = x_1 + \cdots + x_k + y_{k+1} + \cdots + y_n$. We first prove that when the maximum of $\min_{k=0,1,\ldots,n} f(k)$ is achieved, we must have $x_i = y_i$ for all $i$. If not, not losing generality suppose $x_l > y_l$. Now consider $x_i' = x_i$ for all $i \neq l, l+1$, and $x_l' = y_l, x_{l+1}' = x_{l+1} + x_l - y_l$ (omit the latter step if $l = n$). Let $f'(k)$ be the function of $k$ computed based on $x'$ and $y$. By $x_l > y_l$ we have $f(l) > f(l-1)$. Notice that only $f'(l) = f(l-1) < f(l)$ is reduced and for all other $k \neq l$ we have $f(k) = f'(k)$, so the minimum remains the same. Now we have

$$\sum_{i=1}^{n}\sum_{j=i}^{n} x_i' y_j = \sum_{j=1}^{n}\sum_{i=1}^{j} x_i' y_j = \sum_{j=1}^{n} y_j \sum_{i=1}^{j} x_i'$$

$$= \sum_{j=1}^{l-1} y_j \sum_{i=1}^{j} x_i + y_l \sum_{i=1}^{l} x_i' + \sum_{j=l+1}^{n} y_j \sum_{i=1}^{j} x_i$$

$$\leq \sum_{j=1}^{l-1} y_j \sum_{i=1}^{j} x_i + y_l \sum_{i=1}^{l} x_i + \sum_{j=l+1}^{n} y_j \sum_{i=1}^{j} x_i$$

$$\leq t.$$

So there exists a configuration that maximizes $\min_k f(k)$ with $x_i = y_i$ for all $i$. Now suppose $x_i = y_i$ for all $i$. The constraint becomes

$$\sum_{i=1}^{n}\sum_{j=i}^{n} x_i x_j \leq \varepsilon,$$

which is equivalent to

$$\left(\sum_{i=1}^{n} x_i\right)^2 + \sum_{i=1}^{n} x_i^2 \leq 2\varepsilon.$$

By Cauchy-Schwarz inequality we have

$$\sum_{i=1}^{n} x_i^2 \geq \frac{\left(\sum_{i=1}^{n} x_i\right)^2}{n}.$$

So

$$x_1 + \cdots + x_k + y_{k+1} + \cdots + y_n = \sum_{i=1}^{n} x_i \leq \sqrt{\frac{2nt}{n+1}}.$$

It is easy to verify the equality condition. $\qquad\square$

*Proof of Theorem 20.* Not losing generality, suppose $\hat{g}(x) \in [0,1]$; such a assumption is justifiable since any increasing transformation of $\hat{g}$ does not change $R^B(\hat{g})$. So we only need to consider $\mathbb{C}_{\hat{g}} = \{h : h(x) = h_t(x) = \text{sign}(\hat{g}(x) - t), t \in [0,1]\}$. Let $q(u)$ denote the distribution of $\hat{g}(X)$. Let $\xi(u) = q(u)\Pr(h^*(X) = 1|\hat{g}(x) = u)$. So we have

$$\int_0^t \xi(u)du = \Pr(h^*(X) = 1, \hat{g}(X) < t).$$

So the error of $h_t$ with respect to $h^*$ can be expressed as

$$\Pr((\hat{g}(X) - t)g^*(X) < 0) = \Pr(\hat{g}(X) > t, g^*(X) < 0) + \Pr(\hat{g}(X) < t, g^*(X) > 0)$$

$$= \int_0^t \xi(u)du + \int_t^1 (q(u) - \xi(u))du.$$

On the other hand, the comparison error can be expressed as

$$R^B(\hat{g}) = 2\Pr(\hat{g}(X) > \hat{g}(X'), h^*(X) = -1, h^*(X) = 1)$$

$$= \int_0^1 \xi(u) \int_u^1 (q(v) - \xi(v))dudv.$$

Now consider we do this on the grid with step size $1/n$ and let $n \to \infty$; the integral will be the limit value. So, let

$$S_n = \frac{1}{n^2} \sum_{i=1}^{n} \sum_{j=i}^{n} \xi(i/n)(q(j/n) - \xi(j/n)).$$

So

$$\Pr(\hat{g}(X) > g(X'), h^*(X) = -1, h^*(X) = 1) = \lim_{n \to \infty} S_n.$$

Also, let

$$T_n^t = \frac{1}{n} \left( \sum_{i:i/n<t} \xi(i/n) + \sum_{i:i/n>=t} (q(i/n) - \xi(i/n)) \right),$$

so

$$\Pr((\hat{g}(X) - t)g(X) < 0) = \lim_{n \to \infty} T_n^t.$$

Now let $x_i = \frac{1}{n}\xi(i/n), y_i = \frac{1}{n}(q(i/n) - \xi(i/n))$ in Lemma 21, and we have

$$\min_t T_n^t \le \sqrt{\frac{2nS_n}{n+1}}.$$

Note that $\lim_{n \to \infty} 2S_n = R^B(\hat{g}) \le \nu'$ and let $n \to \infty$ on both side, we have

$$\min_t \Pr[(\hat{g}(X) - t)g(X) < 0] \le \sqrt{\nu'}.$$

$\square$