[Reviews · NeurIPS 2017]

Reviewer 1



This paper studies active learning using two kinds of feedbacks together: label feedback and comparison feedback. Its main insight is that to apply the noise pairwise ranking algorithm in (Ailon and Mohri, 2008) to get an order of examples, and then use binary search to infer their labels. It establishes query complexity bounds on the number of labels and number of comparisons, and bounds on noise tolerance on the two feedbacks. It shows that using pairwise comparison can substantially reduce the label complexity in active learning scenarios. I vote for acceptance of this paper since this studies a nice novel problem that has practical motivations, and establishes the first algorithm analyzing noisy comparison feedbacks. Minor comments: 1. It would be better to establish an excess error guarantees instead of closeness to h^* guarantees. (Although assuming reverse Tsybakov noise condition can directly get it, it is still an unnatural condition in my view.) 2. In line 508 it is worth pointing out that |S_i| > k by the assumption of n. This is crucial, since it leads to a requirement of n >= 1/epsilon^{2kappa-1}. 3. In line 462 it is mentioned that (3) is equivalent to Condition 2. In the literature I only see Condition 2 implying (3), but not the other way round. Also, Condition 2 does not require h^* to be Bayes optimal. Could the authors elaborate on this? 4. It would be interesting to get a full tradeoff on the distribution (Z, X, X') and distribution of (X,Y). (On one extreme, if we have uninformative comparison queries, the model degrades to label-based active learning. On the other extreme, if we have an uninformative label queries, Theorem 12 studies its limitations.) (I have read the author feedback - thanks to the authors for clarifying point Q2.)

Reviewer 2



he authors study the active binary classification setting where + The learner can ask the label of an instance. The labelling oracle will provide a noisy answer (with either adversarial noise or Tsybakov) + In addition to the labeling oracle, there is a comparison oracle. The leaner then can ask which of two instances is more likely to be positive. This oracle's answers is also noisy. The authors provide results which show that if the learner uses comparison queries, then it can reduce the label-query complexity. The problem that the authors study seem to be a valid and interesting one. The presentation of the result however should be improved. In the current state, accurately parsing the arguments is hard. Also, it would have been better if a better proof sketch was included in the main paper (to give the reviewer an intuition about the correctness of the results without having to read the supplements). The core idea is to use the usual active-type algorithm, but then replace the label-request queries with a sub-method that uses mostly comparison queries (along with a few label queries). This is possible by first sorting the instances based on comparison queries and then doing binary search (using label queries) to find a good threshold. One thing that that concerning is the assumption on the noise level. On one hand, the authors argue that the major contribution of the paper compared to previous work is is handling noise. However, their positive result is not applicable when the amount of noise is more than some threshold (which depends on epsilon and even delta). Also, the authors do not discuss if the learner is allowed to perform repeated comparison queries about the same pair of instances. This seems to be helpful. [Furthermore, the authors assume that the outcome of the queries are iid, even when they have been asked about overlapping pairs]

Reviewer 3



This paper considers a new active learning setting, where in addition to a labeling oracle, the learner has access to a comparison oracle. It gives lower bounds, and a general algorithm with upper bounds under various noise conditions. In my opinion, this paper is well-motivated, clear, and contains sufficient contribution for NIPS. It considers a novel interactive learning setting, and gives an interesting result that with comparison oracle, the d and 1/epsilon factors can be separated in the total query complexity. Its reduction from learning with comparison oracle to learning 1-dimensional threshold is also quite nice. However, one (minor) technical mistake appears many times in the paper: the dependency of epsilon and kappa in the label complexity under Tsybakov noise condition (under their notation) should be (1/epsilon)^(2-2/kappa). Some other minor issues I have are: 1. The term "distribution-free" in line 177 is confusing to me. In my opinion, the algorithm is depends on the distribution, since it needs to know the Tsybakov noise parameter kappa. 2. In the presentation of the theorems, the SC_comp is the expectation, but SC_label is a high probability bound. It would be better to state SC_comp and SC_label in the same manner. 3. In line 225-231, the parameters M, tau, kappa (here kappa is overloaded), etc. are not defined in the main text. I understand this is due to space limit, but it would be better if the authors could find some alternative way to present their result for learning halfspaces. --- After reading author feedback, I agree with the authors that my claim about the label complexity was incorrect. I didn't notice they were using P(h \ne h^*) in the definition of the label complexity.